# Impacts of rapid mass vaccination against SARS-CoV2 in an early variant of concern hotspot

Jörg Paetzold [1,6 ✉], Janine Kimpel [2], Katie Bates[3], Michael Hummer[4], Florian Krammer [5], Dorothee von Laer [2] & Hannes Winner [1,6]

We study the real-life effect of an unprecedented rapid mass vaccination campaign. Following a large outbreak of the Beta variant in the district of Schwaz/Austria, 100,000 doses of BNT162b2 (Pfizer/BioNTech) were procured to mass vaccinate the entire adult population of the district between the 11th and 16th of March 2021. This made the district the first widely inoculated region in Europe. We examine the effect of this campaign on the number of infections, cases of variants of concern, hospital and ICU admissions. We compare Schwaz with (i) a control group of highly similar districts, and (ii) with populations residing in municipalities along the border of Schwaz which were just excluded from the campaign. We find large and significant decreases for all outcomes after the campaign. Our results suggest that rapid mass vaccination is an effective tool to curb the spread of SARS-CoV-2.

[1] University of Salzburg, Department of Economics, Residenzplatz 9, A-5010 Salzburg, Austria. [2] Institute of Virology, Department of Hygiene, Microbiology and Public Health, Medical University of Innsbruck, Peter-Mayr-Str. 4b, 6020 Innsbruck, Austria. [3] Department of Medical Statistics, Informatics and Health Economics, Medical University of Innsbruck, Innsbruck, Austria. [4] The Austrian National Public Health Institute (Gesundheit Österreich GmbH, GÖG), Stubenring 6, 1010 Vienna, Austria. [5] Department of Microbiology, Icahn School of Medicine at Mount Sinai, One Gustave L. Levy PlaceBox 1124 New York, NY 10029, USA. [6] These authors contributed equally: Jörg Paetzold, Hannes Winner. ✉email: Joerg.Paetzold@sbg.ac.at

n the autumn of 2020, the emergence of SARS-CoV-2 variants of concern was detected in Europe and elsewhere[1–3]. By spring 2021, one of the largest outbreaks of Beta and Alpha/E484K in Europe occurred in the district of Schwaz, Austria[4]. In response to this local outbreak, the Government of Austria and BioNTech joined forces in an effort to supply 100,000 extra vaccine doses of BNT162b2 to rapidly mass vaccinate the entire adult population (16+) of Schwaz. Between 11th and 16th of March, more than 70% of the adult population of Schwaz received their first dose of BNT162b2, which made Schwaz the first widely inoculated region in Europe. This stood in sharp contrast to the slow vaccination progress of the rest of the country, which had a vaccination coverage of 10% at that time. Thus, this local mass vaccination campaign created stark differences in vaccine coverage at the district level of otherwise highly integrated regions with similar spread of SARS-CoV-2 prior to the campaign. We exploit this stark difference in local vaccine coverage to study infections, variants of concern (VoCs), hospitalizations and intensive care unit (ICU) admissions following this mass vaccination campaign. This local, population-wide mass vaccination event provides an opportunity to study the impact of rapid vaccination campaigns against SARS-CoV-2 and its major VoCs.

Previous evidence from real-world coronavirus disease 2019 (COVID-19) vaccination campaigns is mostly based on the comparison of groups which were prioritized in national vaccination plans (e.g., older people) with unvaccinated controls[5–9]. Another approach to quantify the impact of real-world COVID-19 vaccinations is to measure the overall effect of the vaccination program on an entire population[10]. In the district of Schwaz, the entire adult population was offered vaccination (and administered within 5 days), regardless of their age or any other factors. This allows us to compare outcomes of a general population living within the same geographical area but across district borders, resulting in very different vaccine coverage. Our study design keeps confounding factors such as the healthcare system, local conditions facilitating the spread of SARS-CoV-2, and general population characteristics as constant as possible. Finally, due to the occurrence of VoCs in the district of Schwaz, our study also provides evidence of the real-life effect of the vaccine regarding variant cases.

## Results

**Impact of the mass vaccination campaign on vaccine coverage.** Figure 1 plots the shares of the vaccinated adult population for the district of Schwaz as well as for all other Tyrolian districts (pooled together). Prior to the first dose of the campaign (11th to 16th of March), vaccination coverage of first doses was approximately 10% in Schwaz and everywhere else. After the first campaign week, vaccination coverage increased to more than 70% of the adult population. The stark difference between Schwaz and the other districts persisted over months, providing a unique setting to study the impact of the vaccine against SARS-CoV-2.

**Schwaz vs. synthetic control group.** To examine the impact of this stark difference in vaccination coverage we first used the daily number of SARS-CoV-2 infections at the district level as the respective outcome variable. We calculated cumulative daily infections from the second week of January 2021 onwards. We employed the synthetic control group method (SC method) which allowed us to estimate what would have happened to Schwaz in the absence of the mass vaccination campaign (see "Methods" section for further details). Figure 2a shows the cumulative daily infections per 100,000 inhabitants for Schwaz and the synthetic control group. Figure 2b depicts the corresponding 7-day incidence (per 100,000) of daily infections as the

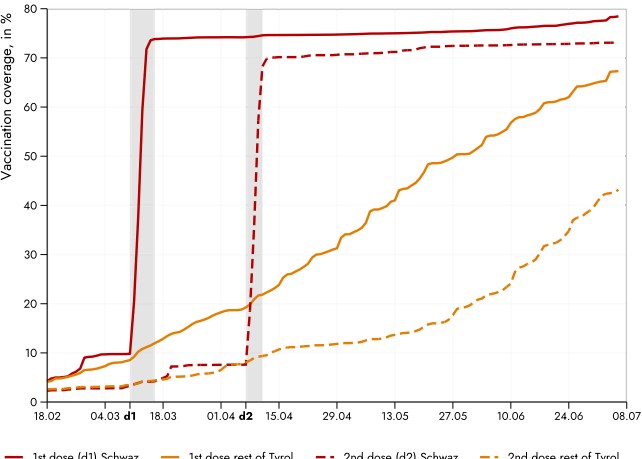

**Fig. 1 Vaccination coverage of adult population in Schwaz and the rest of Tyrol.** The figure displays the shares of the adult population that received the first (solid line) and second dose (dashed line) of vaccination, respectively. Schwaz is plotted in red, while the other (eight) Tyrolian districts are pooled and their mean depicted in orange. The shaded areas indicate the period of the first (d1: 11th to 16th of March 2021) and the second (d2: 8–11 April 2021) roll-out of mass vaccination.

outcome variable. Two observations stand out: First, the treatment and the (synthetic) control group had very similar spread of SARS-CoV-2 infections prior to the mass vaccination campaign, confirming that the two groups are highly comparable. Second, although infections in Schwaz increased somewhat sharper than in the control group in the first days after the first dose, infection dynamics started to diverge around 2–3 weeks later. This is exactly the time period after which the first effects of BNT162b2 materialized in the original phase 2/3 clinical trial[11]. While Schwaz followed its distinct and substantial decline in cases after the second dose, the control group witnessed a sharp increase with high incidence rates throughout April. These high incidence rates in the control group only began to sink once a general trend of decreasing infection levels across Europe started in spring 2021. Around four months after the first dose we found the cumulative daily infections per 100,000 inhabitants in the control group to be about 2469, and 1510 in Schwaz. We tested for the significance of this difference using a permutation test[12,13], which resulted in a p-value of 0.005, suggesting that the probability of observing the large treatment effect of Schwaz by pure chance is very low (see Supplementary Fig. 2 and the corresponding description of the test). The difference in infections between Schwaz and the synthetic control can be interpreted as an estimate of avoided infections. Our results suggest that the mass vaccination campaign resulted in 959 avoided infections in the four months after the first dose of vaccine. This is equivalent to a reduction of 38.8% compared to the synthetic control. It should be noticed that this estimate cannot be directly compared to individual-level efficacy numbers published in the original clinical trial[14]. Different from a clinical trial, the impact of a vaccination program on an entire population hinges on additional factors such as vaccine coverage, vaccine uptake of subgroups, or suboptimal immune status of individuals in the population (with Schwaz having only partial protection over parts of the study period). Furthermore, the population in our control group received vaccination over time as well, which again is different from the original clinical trial design (see Fig. 1).

We also studied cumulative daily infections per 100,000 inhabitants by age group. Figure 5 in the Supplement depicts the cumulative daily infections per 100,000 inhabitants for each

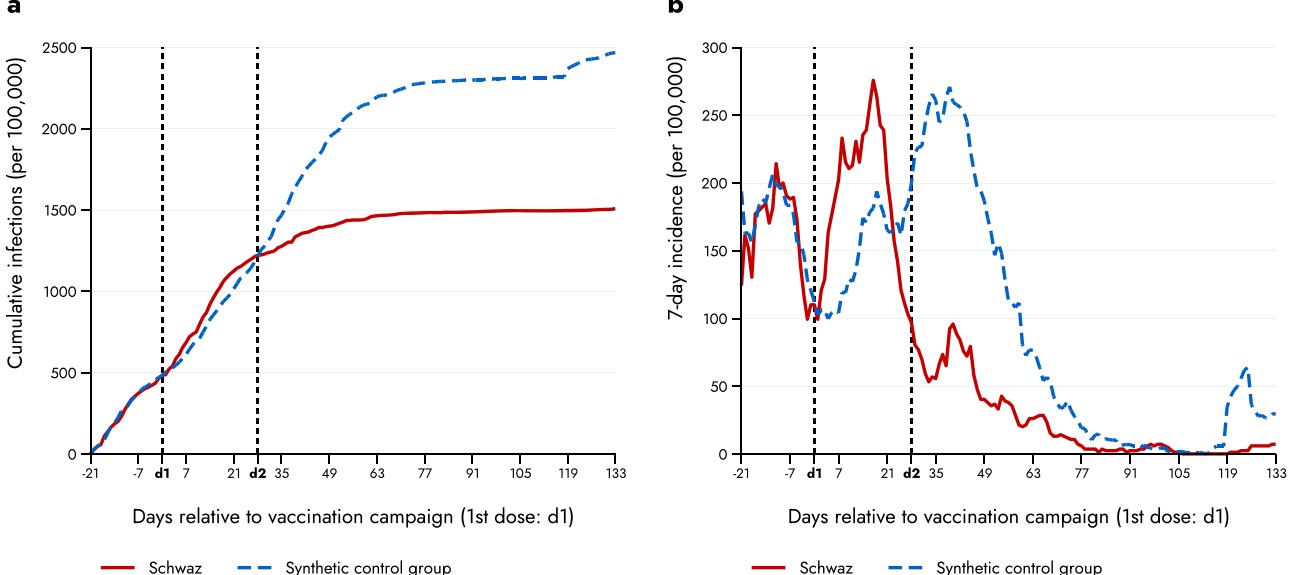

**Fig. 2 Daily infections of Schwaz versus synthetic control group.** (**a**) depicts cumulative daily infections (per 100,000) for Schwaz (solid red line) and the synthetic control group (dashed blue line). (**b**) shows the 7-day incidence (per 100,000) for Schwaz (solid red line) and the synthetic control group (dashed blue line). The chosen donors include Hartberg-Fürstenfeld (24.1%), Hermagor (10.6%), Liezen (0.5%), Reutte (63.8%) and Steyr Stadt (1.1%). The horizontal axis indicates the number of days relative to vaccination campaign (dose 1, indicated by "d1"). The pre-treatment period started 21 days (three weeks) before the first dose, the post-treatment period ended 133 days (19 weeks) after the first dose. The vertical dashed lines represent the first dose (d1) and the second dose (d2) administered in the vaccination campaign.

age group separately. It can be seen that cumulative daily infections (per 100,000 inhabitants) for Schwaz and its synthetic control group were the largest in the youngest age group, with 2827 cumulative infections for the control group and 1566 for Schwaz (i.e., 1261 avoided infections). For the oldest age group (80+) we observed 504 cumulative daily infections for Schwaz and 934 in the synthetic control group (i.e., 430 avoided infections). Thus, while the youngest age group experienced the largest number of avoided infections, the relative reduction was comparable across age groups (44.6% = 1261/2827; 46.0% = 430/934).

Next, we examined hospital admissions related to confirmed SARS-CoV-2 infections. For this outcome variable, we only had weekly data up to calendar week 25 available (i.e., 15 weeks after dose 1 of the campaign). Figure 3a shows the cumulative weekly hospital admissions per 100,000 inhabitants for Schwaz and the synthetic control group (Fig. 6a in the Supplement depicts the corresponding non-cumulative weekly hospital admissions). Prior to the mass vaccination campaign, both the treatment and control group had very similar numbers of hospital admissions. Around 4 weeks after the first dose administered during the campaign, the number of hospital admissions started to diverge. 15 weeks after the first dose we found the cumulative weekly hospitalizations per 100,000 inhabitants was 129 in the synthetic control group and 73 in Schwaz. Relating this difference of 56 (avoided) hospitalizations to the 129 hospitalizations in the synthetic control group means a reduction in hospital admissions of about 43.4%. Furthermore, we studied admission to ICUs related to a confirmed SARS-CoV-2 infection. Figure 3b shows the cumulative weekly ICU admissions per 100,000 inhabitants for Schwaz and the synthetic control group (Fig. 6b in the Supplement shows the corresponding non-cumulative weekly ICU admissions). ICU admissions of the two groups started to diverge around 5 weeks after the first dose. 15 weeks after the first dose the cumulative weekly ICU admissions (per 100,000 inhabitants) was 24.4 in the synthetic control group and 20.1 in Schwaz. Relating this difference of 4.3 (avoided) ICU admissions gives a reduction of

around 17.6%. Apart from the fact that ICU incidence rates only include small numbers, the lower effect on ICU admissions may be explained by the high vaccination rate among old-age individuals and risk groups also in the control group (which followed the age gradient of the national vaccination plan).

**Schwaz vs. bordering municipalities**. In addition to the analysis based on the synthetic control group, we also compared the district of Schwaz with adjacent municipalities located along the district border using an event-study approach. Event studies are commonly used in econometrics to assess the effect of an exogenous event on a variable of interest. Our analysis examined infections among local populations living within the same geographic area, but with stark differences in vaccine coverage after the campaign. In this analysis, we also used VoC cases as additional outcome variable, since sequencing data were available only for the state of Tyrol (but not for all districts used in the SC method).

Figure 4 plots the weekly treatment effects of our event-study model, capturing the difference between Schwaz and the border municipalities relative to the reference period (week of the 1st dose). The figure shows the weekly difference-in-difference coefficients with the associated 95%-CI (see Methods section for further details). Figure 4a is based on all infections as the respective outcome variable, whereas Fig. 4b focuses on confirmed cases of the major VoCs (Beta, Alpha/E484K, and Delta). Both panels of the figure show that in the weeks prior to the mass vaccination campaign, the differences between Schwaz and the border municipalities were not statistically different from zero. Starting 3-4 weeks after the first dose, we found that the number of new cases in Schwaz significantly decreased relative to the border municipalities. This is true for both overall infections as well as for the VoCs, although the decrease is somewhat lower for the variant cases. We found the difference between Schwaz and the control group to be the largest in the first weeks after the second dose, and then becomes somewhat smaller over time. This

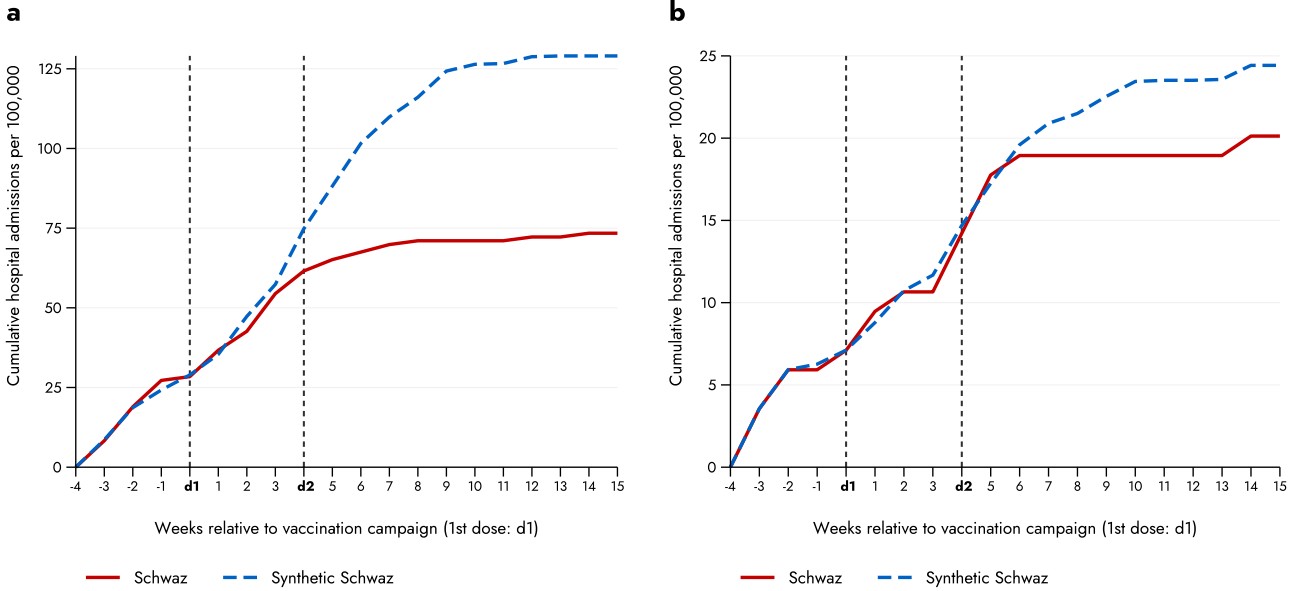

**Fig. 3 Hospital and intensive care unit (ICU) admissions in Schwaz versus synthetic control group.** The figure shows the cumulative weekly hospital admissions (per 100,000) related to a confirmed SARS-CoV-2 infection for Schwaz and the synthetic control group. (**a**) relates to general hospital admissions, and (**b**) to the ones in intensive care units (ICUs). The chose donors include Grieskirchen (15.8%), Reutte (61.9%) and Wiener Neustadt Land (22.3%) in the case of general hospital admissions, and Bregenz (7.8%), Bruck-Mürzuschlag (20.5%), Hermagor (11.3%), Neusiedl am See (51.5%) and St. Pölten (8.9%) for ICU admissions. The horizontal axis shows the number of weeks relative to vaccination campaign (dose 1). The pre-treatment period started four weeks before the first dose, the post-treatment period ended 15 weeks after the first dose. The vertical dashed lines represent the first dose (d1) and the second dose (d2) administered as part of the mass vaccination campaign.

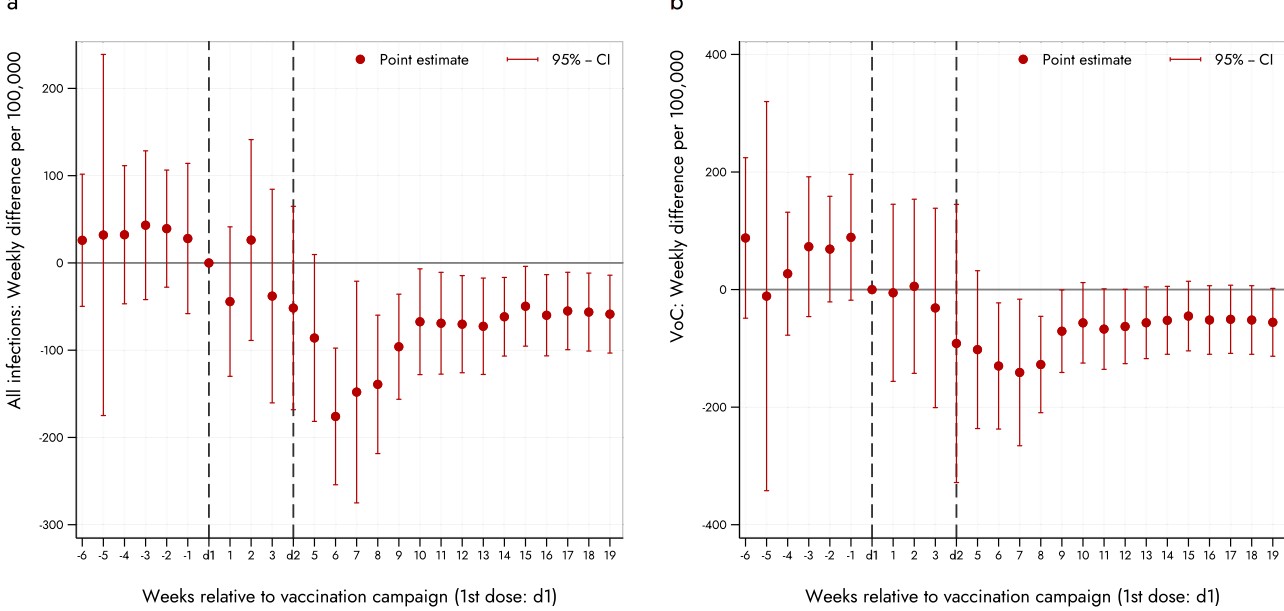

**Fig. 4 Difference in weekly infections of SARS-CoV-2 and its variants of concern (VoCs) in Schwaz and the neighboring municipalities.** The figure displays the results from regression Eq. (1) using weekly cases (per 100,000) as outcome variable for Schwaz and its bordering municipalities. (**a**) refers to all infections, and (**b**) to the sum of variants Beta, Alpha/E484K and Delta. The plotted points are the weekly coefficients $\beta_k$ which represent the mean difference in weekly cases between Schwaz and the border municipalities relative to the reference week d1 (conditional on municipality- and week-fixed effects). The point coefficient for each week is shown together with the 95%-confidence interval. The horizontal axis displays the number of weeks relative to vaccination campaign (dose 1). The vertical dashed lines represent the first dose (d1) and the second dose (d2) administered as part of the mass vaccination campaign.

pattern may be caused by the gradual increase in vaccine coverage of the control border municipalities, but we do not have municipality-level vaccine data to directly test for this explanation.

To calculate the overall effect of the vaccination campaign compared to the neighboring border municipalities, we ran standard two-period DID estimations as shown in Eq. (2) of the Methods Section. These estimates represent the average post-

treatment effect of the weekly coefficients after the vaccination campaign depicted in Fig. 4a. 19 weeks after the roll-out of the first dose, the average post-campaign effect shows a reduction by about 101.5 weekly infections per 100,000 in Schwaz relative to the border municipalities (see column 1 of Table 3 in the Supplement). To determine the percent reduction in new infections due to the vaccination campaign (relative to the border municipalities) we used a log-transformed regression and obtained semi-elasticities (see also the Methods section). We found a significant reduction of the average post-treatment DID in weekly new infections of around −71.1% (95% CI: −85.8 to −41.2).

When using VoCs as a respective outcome, we found a statistically significant average post-campaign reduction of around 114.4 cases per week in Schwaz relative to the border municipalities (see column 2 of Table 3 in the Supplement). The corresponding semi-elasticity indicates a reduction of 78.8% (95%-CI: −92.7 to −38.5). Columns 3 and 4 of Supplementary Table 3 are based on hospital admissions related to a confirmed SARS-CoV-2 infection (per 100,000 inhabitants) as outcome variable for the DID. For general hospital admissions, we found a statistically significant weekly reduction of −6.3 in Schwaz relative to the municipalities in the neighboring districts. The corresponding semi-elasticity indicates a reduction of 33.6% (95%-CI: −48.9 to −13.7). In case of weekly ICU admissions related to a confirmed SARS-CoV-2 infection (per 100,000 inhabitants) we estimated a statistically significant reduction of 4.1 ICU admissions per week in Schwaz relative to the border municipalities, which translates into a semi-elasticity of −20.5% (95%-CI: −31.2% to −8.1%). Due to the small number of hospitalizations in the neighboring border municipalities, the comparison of hospital admissions and ICU is based on weekly observations from all municipalities of the two neighboring districts (Kufstein and Innsbruck-Land).

Table 1 summarizes the percent reductions we calculated above for the event study (two-period DID) and the synthetic control method, respectively. Comparing the numbers shown in the table suggests that the reductions in the respective outcomes are often of similar magnitude, independent of the method we used. It should be noticed, however, that the numbers are based on different control regions, namely the border municipalities (in case of the event study) and the control districts (in case of the synthetic control group). Furthermore, we do not have sequencing data on variants from districts outside of Tyrol and thus, cannot use this data for the SC method.

## Discussion

This retrospective observational study examines COVID-19 vaccine effectiveness at the population level in the district of Schwaz, an early VoC hotspot that became one of the first highly vaccinated regions in Europe. Our analysis uses a control group of districts highly similar to Schwaz regarding many population characteristics. Further, we zoom in on border municipalities residing just outside of the treated district.

Our analysis reveals that the mass vaccination campaign is associated with a significant reduction in new SARS-CoV-2 infections of around 40% relative to control districts, and around 70% when using bordering municipalities for comparison. We find similar significant reductions in variant cases. It is important to note that the dominant variant in Austria at the time of our study was Alpha, and that the Delta variant entered Austria only at the end of our study period. Our analysis also shows a significant reduction in hospital as well as ICU admissions associated with SARS-CoV-2. Overall, our results suggest that the rapid mass vaccination campaign was successful in curbing the spread of the virus, including its main VoCs at that time.

A limitation of our study is that it is not a randomized clinical trial but an observational study, which may be influenced by confounders such as lockdown policies or behavioral changes. While almost all non-pharmaceutical interventions (such as school measures, or curfew restrictions) were identical for Schwaz and the different control groups, there was an additional SARS-CoV-2 test requirement between the 11th of March and the 8th of April when crossing the border of the district. We investigate for every Austrian district with the same test requirement (in total five other districts) whether infections dropped in a similar magnitude as they did in Schwaz. None of the five districts experienced a decline in any comparable way after the test requirement (see Fig. 8 in the Supplement). Finally, by announcing the mass vaccination campaign a signal may have been sent to the population of Schwaz that the situation was serious, which could have led to behavioral changes. It is important to note that the mass vaccination campaign was announced on the 3rd of March, many weeks before the significant drop in cases after the second dose occurred (8–11 April). We analyze Google mobility data and find no reduction in mobility in Schwaz relative to the synthetic control group, which would be a sign of changes in people's behavior (see Fig. 7 in the Supplement). Taking together, our findings suggest that the large reduction of infections in Schwaz was driven by the mass vaccination campaign.

Our estimates of the percent reduction in cases following the vaccination campaign cannot be directly compared to individual-level efficacy numbers published in the original clinical trial[11]. Different from a clinical trial, the impact of a vaccination program on an entire population hinges on additional factors such as vaccine coverage, vaccine uptake of subgroups, behavioral differences, or suboptimal immune status of individuals in the population, potentially affecting the external validity of our study. Nevertheless, given that the district of Schwaz was one of the first widely inoculated regions worldwide, we believe that our results are of large interest to other global regions. Our results suggest that rapid population-wide mass vaccination can be an effective tool to reduce overall infections as well as help to curb the spread of VoCs. This will be especially important when vaccines become more easily available at a large scale by the end of 2021[14].

**Table 1 Summary table on the percent reduction in new infections as well as hospitalizations using the event-study and the synthetic control method.**

| Method | New infections | variants of concern (VoCs) | Hospitalization | |
|---|---|---|---|---|
| | | | General admissions | intensive care unit (ICU) |
| Event Study (DID) | −71.1% [−85.8%;−41.2%] | −78.8% [−92.7%;−38.5%] | −33.6% [−48.9%; −13.7%] | −20.5% [−31.2%;−8.1%] |
| Synthetic Control | −38.8% | | −43.4% | −17.6% |

For the SC method, we related the number of avoided cases to the number of observed cases in the synthetic control group. For the two-period DID, we used a log-transformed regression to calculate a semi-elasticity which is then transformed into relative effects. See "Methods" section for details. 95%-CI are reported in brackets.

## Methods

**Data sources.** For our retrospective observational study, we used data from the Austrian epidemiological reporting system (Österreichisches Epidemiologisches Meldesystem, EMS). These data are collected by the Austrian National Public Health Institute (Gesundheit Österreich GmbH, GÖG), along with information on hospital admissions due to COVID-19 diagnosis. Our database comprises municipality/district-level epidemiological data from the universe of all Austrian districts, and all municipalities within those districts. We employed all infections, VoC cases (i.e., Beta, B1.1.7/E484K and Delta), hospital and ICU admissions recorded for those geographical units. Sequencing as well as vaccination data is only available for the state of Tyrol, which responded with comprehensive sequencing of almost all SARS-CoV-2 PCR-positive cases after the large outbreaks of Beta and Alpha/E484K in February 2021. The study has been reviewed and approved by the ethics committee of the University of Salzburg.

**Study design and statistical analysis.** Our study design exploited the fact that regions which share many geographical as well as socio-demographic characteristics ended up with very different vaccine coverage following the vaccination campaign. To distinguish the possible effects of the vaccination campaign on (variant) cases and hospitalizations from other factors, we used two complementary approaches.

First, we used the *synthetic control method* (SC), which is widely applied in causal analysis[12,13,15], and also in recent health and Covid-19 research[16,17]. The synthetic control group is constructed as a convex combination of donors (i.e., 92 Austrian districts) by minimizing a weighted sum of squared deviations for the matching variables. Weights are chosen in a data-driven way to approximate as closely as possible the pre-treatment characteristics of Schwaz. As matching variables we used the SARS-CoV-2 infection spread prior to the vaccination campaign and additional covariates such as population size, geographical area size and the number of municipalities within a district. All calculations were executed with the software STATA (MP version 16) using the package Synth, which is based on the R-package Synth[18]. For case numbers, the chosen donors include Hartberg-Fürstenfeld (24.1%), Hermagor (10.6%), Liezen (0.5%), Reutte (63.8%) and Steyr Stadt (1.1%), with none of them being neighboring districts. All other Austrian districts receive zero weight, and identifiability is ensured by the constraint that non-negative weights summing to one. Supplementary Table 1 summarizes further details on the profiles of Schwaz and the synthetic control group. Notice that we choose separate donor districts for each outcome variable to ensure comparable pre-intervention trajectories in each case (the donors are listed in the notes of Figs. 2 and 3). Once the treatment took place (i.e., roll-out of the first dose in the campaign), the respective outcome variable is then compared between Schwaz and its synthetic counterpart. This allows us to estimate what would have happened to Schwaz in the absence of the mass vaccination campaign. To evaluate the significance of the differences observed between Schwaz and the synthetic control group, we employed a permutation test based on an exact Fisher test[12,13]. For this purpose, we executed a placebo test where we applied the SC method sequentially on each of the 92 districts in the donor pool ("placebo units"), using the date of the roll-out of the first dose in Schwaz as the treatment date (see Supplementary Figs. 2, 3). Finally, we checked the robustness of our results using a one-leave out test where we leave out the chosen donors on the baseline control group one at a time (see Supplementary Fig. 4). Both the placebo and the one-leave-out test suggest robust estimation results for the synthetic control method.

Second, we made use of our very fine-grained geographical data to compare Schwaz with adjacent municipalities just outside the border of the district. This ensures that the populations living in these border municipalities share many geographical and socio-demographic characteristics (e.g., local mobility) with Schwaz but were excluded from the mass vaccination campaign. We employed an event-study model based on a *difference-in-difference* (DID) design to measure the impact of the campaign in Schwaz relative to the border municipalities[15,19]. We selected those border municipalities on the basis of road connectivity to the district of Schwaz. Specifically, we only selected border municipalities outside the district of Schwaz as control units when there existed a direct road link between the respective border municipality and Schwaz. This ensures that the populations living in these border municipalities share many geographical, institutional (e.g., healthcare system) and socio-demographic characteristics (e.g., local mobility) with Schwaz but were excluded from the mass vaccination campaign. Supplementary Table 2 provides a balancing table on selected characteristics of the chosen municipalities in Schwaz and the ones along the neighboring districts.

We estimated a two-way fixed effects model including an indicator variable for municipalities located in Schwaz as the treated units. We started from the 18th of January 2021 and estimated for each week $k$ the DID of new infections (per 100,000 inhabitants) between the group of bordering municipalities and the ones of Schwaz. The regression equation is given by

$$y_{i,w} = \delta_i + \delta_w + \sum_{k=-6}^{-1} \beta_k D_{i,w+k} + \sum_{k=1}^{19} \beta_k D_{i,w+k} + \epsilon_{i,w} \qquad (1)$$

where $y_{i,w}$ denotes the weekly sum of new infections (per 100,000) in municipality $i$ (Schwaz or border municipalities) at week $w$. $\delta_i$ and $\delta_w$ denote municipality- and week-fixed effects, and $D_{i,w} = 1$ for weeks after the treatment and if a municipality

received a treatment (i.e., is located in Schwaz). $k$ in the sum operators indicate pre- (first sum) and post- (second sum) treatment effects. $\varepsilon_{i,w}$ is a classical i.i.d. error term. Standard errors are clustered at the municipality level. Our coefficients of interest are the $\beta_k$, which measure the difference in the outcome variable (e.g., daily infections) between Schwaz and the neighboring border municipalities at a given week $k$ relative to the omitted reference category, which is the week of the first dose of the campaign (11th to 16th of March).

Estimating (1) is a common way in econometrics to estimate the dynamic effects of an intervention (here vaccination campaign) on an outcome variable (in our case infections and hospitalizations)[15,19]. It allows us to draw a graph (Fig. 4) showing the pattern of the impact of vaccination. To calculate the overall (static) causal effect of the vaccination campaign in Schwaz relative to the neighboring border municipalities, we further employed a standard two-period DID analysis (before–after comparison). We estimated one post-treatment effect that comprised the average effect over all post-campaign weeks starting 14 days after the roll-out of the first dose which is approximately the time period after which first effects of BNT162b2 materialized in the original clinical trial[11]. Empirically, we ran separate regressions for each outcome variable (sum of all infections, VoCs, general hospitalizations and ICU admissions) in the form

$$y_{i,w} = \beta_0 + \beta_1 \text{Treat}_i + \beta_2 \text{Post}_w + \beta_3 (\text{Treat}_i * \text{Post}_w) + \epsilon_{i,w} \qquad (2)$$

Using a treatment dummy (with entry one for Schwaz and zero otherwise), an indicator variable for the post-treatment period (i.e., the one after 11th of March) and an interaction term between the treatment dummy and the post-treatment period dummy as independent variables. The interaction term represents the static DID-estimate. To put this estimate into perspective, we also calculated the percentage impact of the treatment effect by re-running regression (2) taking the logarithm of the respective dependent variable (i.e., taking $\ln(y)$ rather than $y$ as the outcome variable). To circumvent the log(0) problem, we used a hyperbolic sine transformation on the outcome variable. Through this we obtained semi-elasticities which can be transformed to percentage effects according to $(e^\beta - 1) \times 100$, where $e$ is Euler's number and $\beta$ is the parameter estimate of the aforementioned interaction term. Thus, the semi-elasticity measures the percentage change in the respective outcome (e.g. number of infections) in response to a change in another variable, in our case the binary interaction term of Eq. (2). We reported the point estimates in Supplementary Table 3, and the percentage effects in Table 1 of the main text. Standard errors were clustered at the municipality level.

Overall, the synthetic control and the event-study (DID) method have their strengths and weaknesses. While the border design of the event study uses only nearby municipalities which share many socio-demographic and geographical characteristics (e.g., local mobility) with Schwaz, potential cross-protection through spillovers from the treated district cannot be ruled out. In contrast, the SC method relies only on districts much further apart from Schwaz, which rules out such cross-protection effects. However, these districts do not share some of the local geographical characteristics that potentially influence infection spread. A comparison of the estimated parameters from both methods indicates the validity of our causal statements and results.

**Reporting summary.** Further information on research design is available in the Nature Research Reporting Summary linked to this article.

## Data availability

For this study we used data from the Austrian epidemiological reporting system (Österreichisches Epidemiologisches Meldesystem, EMS). These data are collected by the Austrian National Public Health Institute (Gesundheit Österreich GmbH, GÖG), and is provided to the researchers through a restricted-access agreement. Access to this dataset can be given to other researchers through direct application for data access to the GÖG (see https://datenplattform-covid.goeg.at/antrag). Sequencing and vaccination data is made available by the "Amt der Tiroler Landesregierung", which can be applied via email (lwz@tirol.gv.at).

## Code availability

Standard epidemiological analyses were conducted using standard commands in STATA/MP 16.1 (ref. 36) and the STATA package Synth. The codes to replicate all the statistical analyses are accessible using the following URL: https://github.com/hwin365/2021_schwaz[20].

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

## Acknowledgements

The authors are grateful to Daniela Schmid and Lukas Richter from AGES for providing SARS-CoV-2 qPCR data and critical discussion. We also would like to thank Elmar Rizzoli and Thomas Geiler from Amt der Tiroler Landesregierung for providing sequencing and vaccination data for the state of Tyrol. We also thank the GÖG for data assistance. We are grateful for seeing the support of Katie Bates through the FWF Austrian Science Fund Lise Meitner Award [M-3069-B].

## Author contributions

J.P. conceived and codesigned the study, performed statistical analyses and cowrote the first draft of the article. H.W. conceived and codesigned the study, performed the statistical analyses and cowrote the first draft of the article. J.K., F.K., and D.V.L. codesigned the study. K.B. cowrote the draft of the article. J.P. and H.W. equally contributed to data collection and acquisition, as well as database development. M.H. provided data preparation and management. All authors contributed to the discussion and interpretation of the results, and to the writing of the manuscript. All authors have read and approved the final manuscript.

## Competing interests

The Icahn School of Medicine at Mount Sinai has filed patent applications relating to SARS-CoV-2 serological assays and NDV-based SARS-CoV-2 vaccines which list Florian Krammer as co-inventor. Mount Sinai has spun out a company, Kantaro, to market serological tests for SARS-CoV-2. Florian Krammer has consulted for Merck and Pfizer (before 2020), and is currently consulting for Pfizer, Seqirus, and Avimex. The Krammer laboratory is also collaborating with Pfizer on animal models of SARS-CoV-2. The funders had no role in the design of the study; in the collection or analyses of data, in the writing of the manuscript, or influence on the judgments and actions with regard to objective data presentation and interpretation. Thus, there was no threat to the objectivity, integrity, and value of a publication. For all other authors, no competing interests exist.
