## [Peer Review File · Nature Communications]

Impacts of rapid mass vaccination against SARS-CoV2 in an early variant of concern hotspotREVIEWER COMMENTS

Reviewer #1 (Remarks to the Author):

Thanks for the opportunity to review this work. The manuscript is extremely relevant, overall and in this specific historical moment. I shall also mention that it was particularly well crafted and pleasant to read.

The authors exploit what appears to be a favourable quasi-experimental setting to evaluate the real-world population-level effectiveness of a mass vaccination campaign on three main outcomes: new infections, hospitalisations, and ICU admissions. The authors find that the mass vaccination campaign in the district of Schwaz had a strong effect in curbing the spread of the disease and its negative health system effects. The authors propose two complementary approaches to estimate the effect of the mass vaccination campaign, a synthetic control (SC) method and a DID approach. The two estimation approaches have roughly comparable results for infections (about 60% reduction) and ICU admissions (about 21% reduction), whilst estimates for hospitalisation differ slightly (about 76% for SCM, about 35% for DID).

The authors are cautious in presenting their results, clearly refraining from comparing them to clinical trial individual-level vaccine effectiveness results. Yet, this kind of real-world population-level studies are very much needed at this stage of the pandemic.

The authors are also transparent and accurate in describing their empirical approach, which is well designed and strong.

However, given the importance of the topic addressed by the manuscript, I'm afraid to report that a few points need to be addressed in order to convince myself and the potential readers of the validity of the results.

Main comments:

1. Spillovers and violation of SUTVA: almost by definition, vaccines have positive externalities on non-treated individuals. The protection offered by vaccines – in this case mainly in terms of reduced risk of infection – extends to non-vaccinated people that interact with those that are immunized. This seems relatively likely to happen even if restrictions are in place (families gathering, workers crossing district borders, other economic interactions, etc.). Additionally, from the current text it is not clear enough how the mass vaccination campaign was actually limited to citizens of Schwaz district (i.e. how did the authorities avoid neighbours sneaking in and getting a vaccine?). Consistency of both SC and DID requires the absence of spillovers from treated to non-treated units. However, the authors do not seem to address this problem explicitly in the current version. More specifically, there is neither a discussion of how spillovers are accounted for or prevented by the quasi-experimental setting, nor an attempt to limit them by design. This problem may be relaxed by the SC approach (e.g. excluding neighbouring districts from the donor pool – if this was done, I didn't see it mentioned). However, for the DID design, this problem may seriously jeopardize the credibility of the results.

2. Treatment assignment: Apparently, the treatment assignment (i.e. the decision to implement a mass vaccination campaign) was decided based on one of the outcomes measured (infections). However, this makes the treatment and the control group intrinsically different. Do the authors confirm that this was the rationale for introducing the mass vaccination campaign? Can the authors report (in Appendix) how different the infection rates were in Schwaz compared to neighbouring and other regions in the weeks prior to the launch of the campaign? If that was the rationale for the mass vaccination campaign, does the parallel trends assumption (required for the DID identification strategy) hold? Beware that in this situation parallel pre-trends cannot be considered enough. SC could in principle overcome this limitation by producing a control with similar levels of infections. Box a in Figure 2 casts some doubts about this as vaccinations seem to accelerate and outcomes seem to decelerate with some anticipation in Schwaz, but overall one may still be convinced with some more discussion.

If - instead - the rationale for the mass vaccination wasn't a clear difference in cases in Schwaz, what else was different about Schwaz?

3. When commenting on the results of the DID approach, the authors spend a few words

commenting on the potential role of increasing vaccine uptake in control districts (p 5, lines 226-233). This seems a relevant issue to explore more. If I am not mistaken, both the two-period DID and the SC approaches estimate average effects for the entire post period. However, TWFE results seem to suggest that effects are driven by the first weeks. Can the authors try to say something more about the heterogeneity of effects across rates of vaccination uptake in control districts? Perhaps with a post-period vaccine uptake stratification of the SC analysis? Also: what is the rationale for excluding the vaccination rate from the weighting procedure in SC?

4. Besides suggesting a recent comprehensive paper on synthetic controls methods by Abadie (2021, *Journal of Economic Literature*, <https://doi.org/10.1257/jel.20191450>), I believe the study would benefit in terms of transparency if the following details/robustness checks were reported in the appendix:

- a. Show weighting of other districts composing the SC
- b. Test robustness of results to exclusion of some districts from the SC (see Abadie, 2021)
- c. Show (parallel?) pre-trends between treatment and control group for DID design (neighbouring districts)
- d. Show a full balance table of pre-treatment characteristics (controls vs. Schwaz) for both SC and DID (balance tables along the main set of district characteristics could go in the main text)

Minor comments:

- Reporting the SC estimates in Table 1, alongside the two-period DID, would make the paper easier to read.
- Although the authors make the observational nature of the study clear, it would help to know something about how people were recruited for vaccination.
- The authors may want to consider contrasting their estimates of the effect of a mass vaccination campaign with comparable studies on the effectiveness of other non-pharmaceutical public health interventions, for example, lockdowns.
- The discussion should highlight better the limited external validity of the results to other different settings

Review of Paetzold et al, Nature Communications

September 24, 2021

This is a very interesting paper which in principle I would like to recommend for publication in Nature Communications. The authors seized what they rightly call a “unique opportunity to study the impact of rapid vaccination campaigns against SARS-CoV-2 and its VoCs.” I am sure that this paper will be of interest to a large audience across scientific disciplines. This recommendation, however, is subject to the correction of some careless mistakes in reported numbers as well as improvements on the methodological clarity and reproducibility.

The comments below are unlikely to change the general results and conclusions of the paper, but I hope that they will help make certain aspects clearer and more convincing.

1. I would like to see a descriptive plot of the epidemic curve on the incidence in addition to the one on the cumulative scale. This could replace panel (b) of Figure 2 which shows information already present in panel (a) of the same figure. For Fig 4 it could be added to the supplement.
2. I have only superficial knowledge on the sythetic control method, but to me it does not become clear how the control was actually constructed. I am sure there are numerous ways of doing this and the authors should clearly state what was done.
 - Please specify which mesaure of similarity between Schwaz and the weighted control is being optimized. From doing a bit of research I concluded that it is likely a weighted sum of quadratic deviations across the covariates used for matching. How were these weights chosen? As the different covariates live on very different scales and orders of magnitude this seems like an important question.
 - From the GitHub repository I can see that an implementation in Stata (apparently with default settings) was used to create the sythetic control. Please reference this in the text and provide a reference to the exact underlying method.
3. I do not understand regression equation (1) nor which data were included to fit the model.
 - Why are both of the time indices t and w needed? I think only w should be used. The fact that t is also used makes me suspect that the authors fitted the model using all seven 7-day moving averages of each week. I don't think this makes sense as these observations are not independent and essentially each independent data point will be used seven times. This will in turn lead to too small uncertainty intervals. Only one 7-day value per week should be included. Alternatively, seven one-day values could be included, but these are likely too noisy.
 - How can the term $D_{it,w}$ enter 22 times as a covariate? I assume that something went wrong in the notation as this cannot be what was actually done. As is, equation (1) could be re-written as

$$y_{it,w} = \delta_i + \delta_w + \left(\sum_{k=-6}^{-1} \beta_k + \sum_{k=1}^{16} \beta_k \right) \times D_{it,w},$$

which is not meaningful. My understanding from looking at the code (I am not a Stata user) and

the verbal description is that the authors did something like

$$y_{i,w} = \delta_i + \delta_w + \sum_{k=-6}^{-1} \beta_k D_{i,w-k} + \sum_{k=1}^{16} \beta_k D_{i,w-k},$$

where $D_{i,w-k} = 1$ if and only if (i is a municipality which received the treatment and $w - k = 0$), i.e. the observation comes from Schwaz and is from k weeks after the start of the intervention (which is at week 0).

4. A general question which concerns both equation (1) and the difference in difference approach: I wonder whether this “linear” approach is the most meaningful given the exponential nature of an epidemic. As an example: Consider two otherwise comparable regions A and B which at time 1 have seven-day incidences of 50 and 75, respectively. After an intervention in region B, at time 2 they have 100 and 130, respectively. In your DID setting, the effect of the intervention would be quantified as $(130 - 75) - (100 - 50) = 5$, i.e. a slight worsening of the situation. I would, however, argue that the intervention had a desirable impact given that cases only rose by 73% rather than 100% as in region A. I suggest to re-run the regression model and difference in difference computation using log-transformed incidence values of $z_{iw} = \log(y_{i,w} + 1)$, so that the exp-transformed parameters β_k have a multiplicative interpretation. I assume results to be robust to this, but I think it would make sense to report both versions (one could be moved to the supplement).
5. In Figure 3 I find it difficult to compare the reductions in the different age groups as I am unaware of the general levels of incidence in the different age groups. Maybe the differences look bigger for some age groups because the numbers are generally larger in that age group. I suggest to split the figure into two panels, one showing the current figure and one showing the relative difference in *cumulative incidence since the intervention* (i.e. ignoring everything that happened prior to the intervention).
6. Please provide a reference for the “standard permutation test” mentioned in several instances.
7. I am puzzled by the percentage numbers reported for the reduction of cases, hospitalizations etc on page 5
 - Line 195: 900 avoided infections compared to 2400 total infections is a 37% reduction. 900 avoided infections compared to roughly 1200 total infections since the start of the treatment is a 75% reduction. Where do the 53.6% come from? What am I missing?
 - Line 225: 55.7 avoided hospitalizations for 126.8 in the untreated group is a reduction of 43%. 55.7 is 78% of 71.0 (i.e. the number in the treated group). So it would be correct to say that in the unvaccinated group hospitalization was 78% higher, but the current statement is incorrect.
 - The same problem is present in line 232 (5.2 is 31% of 16.6, not 21.8).
 - Please carefully check *all* reported percentage values and be sure that they really correspond to the verbal interpretation provided!
8. I have some questions with respect to confounding.
 - The setting in Schwaz is probably as close to an experimental design as one can get in an epidemiological study of mass vaccination. However, as the authors correctly point out, this remains an observational study with a risk of confounding. A possible difficulty I see is the following: By announcing the mass vaccination programme, a very strong signal was sent to the population that the situation was drastic and possibly out of hand. Could that have led to adaptations in behaviour which decreased transmission?
 - What made me wonder about this is the fact in the left panel of Fig 5 a rather pronounced effect (though not “statistically significant”) is visible already one week after the intervention. According to e.g. the Australian Department of Health, “*partial protection against COVID-19 may be as soon as 12 days after the first dose, this protection is likely to be short lived*” (<https://www.health.gov.au/initiatives-and-programs/covid-19-vaccines/is-it-true/is-it-true-how-long-does-it-take>

I am not an immunologist, but would not expect to see an effect one week after the first vaccination. Of course one should not overinterpret estimates with wide error bars, but I would still like to ask the authors for a brief comment on this aspect.

- That being said, I do not think that changes in population behaviour are relevant for the observed changes in the medium term, which are the most relevant for the overall results and interpretation.
 - Another difference between Schwaz and the synthetic control (or any point of comparison in the study) is that Schwaz had “one of the largest outbreaks of B.1.351 and B.1.1.7/E484K in Europe”. If at all, this should bias results towards a lower effect of mass vaccination, so this is not problematic. But I would still be interested in a brief comment on whether this outbreak must be assumed to have a relevant impact on the subsequent course of the epidemic in Schwaz.
9. Supplementary Figure 1 does not seem to be referenced in the main text. I think this is an interesting figure as it illustrates that the synthetic control approach works “well enough” and the observed effect is unlikely to be due to chance alone.
 10. The GitHub repository referenced by the authors contains analysis codes, but no data to replicate the presented analyses. I strongly encourage making all materials available in a self-contained and stable repository so the results can be replicated.

Minor comments

1. Please add the word “cumulative” to the axis labels in Fig 4.
2. I suggest to use different colours for curves referring to the whole of Tyrol and to the synthetic control (these are currently both shown in blue)

Comments on the Revision of Manuscript NCOMMS-21-33616

The effects of rapid mass vaccination against SARS-CoV-2 and its Variants-of-Concern: Evidence from an early VoC hotspot

We thank the Editor for inviting us to resubmit the paper in a revised form. We have received very helpful remarks from the two referees (henceforth R1 & R2), which we took very seriously and leading us to substantially revise the paper. We show all changes in the revised manuscript with track changes. Revised/replaced figures are still visible in the manuscript but crossed out by a horizontal red line.

We have done our best to address all suggestions and remarks, and we hope that our responses are sufficiently clear and comprehensible. Below, we address all the concerns and recommendations of the referees as they appear in the reports. All page numbers in our response letter refer to the revised version of the paper.

1 Response to Referee (R1)

We are grateful for your excellent comments and suggestions. Below, we describe step-by-step how we have incorporated them.

1.1 Main Comments

Comment R1.1: *Spillovers and violation of SUTVA: almost by definition, vaccines have positive externalities on non-treated individuals. The protection offered by vaccines – in this case mainly in terms of reduced risk of infection – extends to non-vaccinated people that interact with those that are immunized. This seems relatively likely to happen even if restrictions are in place (families gathering, workers crossing district borders, other economic interactions, etc.). Additionally, from the current text it is not clear enough how the mass vaccination campaign was actually limited to citizens of Schwaz district (i.e. how did the authorities avoid neighbours sneaking in and getting a vaccine?). Consistency of both SC and DID requires the absence of spillovers from treated to non-treated units. However, the authors do not seem to address this problem explicitly in the current version. More specifically, there is neither a discussion of how spillovers are accounted for or prevented by the quasi-experimental setting, nor an attempt to limit them by design. This problem may be relaxed by the SC approach (e.g. excluding neighbouring districts from the donor pool – if this was done, I didn't see it mentioned). However, for the DID design, this problem may seriously jeopardize the credibility of the results.*

The vaccination campaign in Schwaz was indeed limited to residents of the district of Schwaz. To be more precise, every citizen at the age of 16 or above who had his or her main legal residence in the district *prior* the announcement of the campaign (3rd of March 2021) was invited to get vaccinated. This residence requirement was controlled and enforced at the vaccination centres where residents

received their doses. Only in the case of opened but unused jabs at the end of the campaign weekend, residents from towns outside of Schwaz were asked if they wanted to receive the vaccine. We do not have data to examine the exact extent of this practice, but anecdotal evidence as well as **Figure 1** from the manuscript suggest that this effect wasn't large (unfortunately, vaccine coverage rates on municipality level are not available for March 2021).

Regarding the question of protection spillovers to non-vaccinated people that interact with those that are immunized: We cannot rule out that such cross-protection to the mostly unvaccinated population living just outside the district of Schwaz may have occurred. As mentioned in your report, this problem should be much less pronounced in the synthetic control (SC) approach as long as neighbouring districts are not included in the control group. In our case, the neighbouring districts are included in the donor pool but are not selected by the SC algorithm. The chosen control districts are Hartberg-Fürstenfeld with 24.1%, Hermagor with 10.6%, Liezen with 0.5%, Reutte with 63.8% and Steyr Stadt with 1.1%; see also Table S1 in the Supplement. None of these districts share a border with Schwaz, and are often several hundred kilometres away. Therefore, we do not expect serious spillover effects in the SC approach.

When using the border design (i.e. the event-study), cross-protection through spillovers may be present. Nevertheless, we still observed a significant reduction in case numbers in Schwaz relative to the control border municipalities. In the end, both methods we used have their pro's and cons: While the border design uses only nearby municipalities which share many socio-demographic and geographical characteristics (e.g., local mobility) with Schwaz, potential cross-protection cannot be ruled out. On the other hand, the SC method relying on districts which are much further apart from Schwaz rules out such cross-protection effects, but does not share some local geographical features which may affect infection spread. This is why we think the use of both methods is a reasonable decision in order to examine the relationship between the mass vaccination campaign and the medical outcomes. We now discuss the strengths and weaknesses of our approaches to identify a possible causal effect of the vaccination campaigns in the Methods section.

Comment R1.2: *Treatment assignment: Apparently, the treatment assignment (i.e. the decision to implement a mass vaccination campaign) was decided based on one of the outcomes measured (infections). However, this makes the treatment and the control group intrinsically different. Do the authors confirm that this was the rationale for introducing the mass vaccination campaign? Can the authors report (in Supplement) how different the infection rates were in Schwaz compared to neighbouring and other regions in the weeks prior to the launch of the campaign?*

If that was the rationale for the mass vaccination campaign, does the parallel trends assumption (required for the DID identification strategy) hold? Beware that in this situation parallel pre-trends cannot be considered enough. SC could in principle overcome this limitation by producing a control with similar levels of infections. Box a in Figure 2 casts some doubts about this as vaccinations seem to accelerate and outcomes seem to decelerate with some anticipation in Schwaz, but overall one may still be convinced with some more discussion.

If - instead - the rationale for the mass vaccination wasn't a clear difference in cases in Schwaz,

what else was different about Schwaz?

The main reason for the unique vaccination campaign was the outbreak of B.1.351 in the district of Schwaz. The concern was that the mutations that this variant carries in its spike protein may make it less susceptible to the immune response induced by vaccines, especially regarding the vaccine of AstraZeneca (an early efficacy study from AstraZeneca in South Africa indicated reduced efficacy, see Shabir et al. (2021)). Since the vaccine of AstraZeneca was the main vaccine product the Austrian government had ordered by that time, the idea of the campaign was to create a "vaccination ring" using BNT162b2 to contain and eventually eliminate the variant in Austria. Thus, the main reason was not extreme levels of infections which led to the vaccination campaign in Schwaz but rather the occurrence of the B.1.351 variant. In fact, **Figure R.1-1 a** shows that Schwaz was not a unique outlier with regard to infection rates in winter 2021, and that incidence rates in Schwaz in the weeks prior to the campaign did not deviate substantially from the overall average across all other Austrian districts.

Furthermore, **Figure R.1-1 b** shows that there were no substantial differences in the incidence rates between Schwaz and the municipalities located just outside along the border, which we used as controls in our event-study approach. More importantly, they moved in parallel trends many weeks before the campaign, which is an identifying assumption of the DID estimator. Notice that this is also reflected in **Figure 4** of the manuscript, where we report insignificant differences in the pre-trends of infection rates between Schwaz and the border municipalities.

Overall, we think that the common trend assumption is not violated in our application. In the manuscript, both our event-study graph (e.g. **Figure 4**) as well as the Figures based on the SC method (e.g. **Figure 2**) show that in the 6-9 weeks prior dose 2 of the campaign, infection rates were very similar between Schwaz and the respective control regions. In addition, we now included **Figure R.1-1** as **Figure S1** in the Supplement.

Comment R1.3: *When commenting on the results of the DID approach, the authors spend a few words commenting on the potential role of increasing vaccine uptake in control districts (p 5, lines 226-233). This seems a relevant issue to explore more. If I am not mistaken, both the two-period DID and the SC approaches estimate average effects for the entire post period. However, TWFE results seem to suggest that effects are driven by the first weeks. Can the authors try to say something more about the heterogeneity of effects across rates of vaccination uptake in control districts? Perhaps with a post-period vaccine uptake stratification of the SC analysis? Also: what is the rationale for excluding the vaccination rate from the weighting procedure in SC?*

We fully agree that including vaccination data would be interesting to estimate the effects of the vaccination campaign more precisely. Unfortunately, this is not possible because there is no daily or even weekly vaccination data available before the vaccination campaign in March 2021 for districts outside of Tyrol. To produce **Figure 1**, we received special data from the Tyrolean authorities, which is not available for the rest of Austria. Notice that 4 out of the 5 districts used in the SC method as control districts are outside of Tyrol and, therefore, we are not able to include vaccination data in the SC analysis. The same holds true for the ES analysis (i.e., the border

Figure R.1-1: 7-day incidence in treatment- and control group before the vaccination campaign

Notes: The figure shows the 7-day incidence (per 100,000) in the pre-treatment period starting on the 1st of January and ending on the 12th of March 2021 (one day after the first dose of the vaccination campaign in Schwaz was administered). Panel **a** shows the pre-trend for the Austrian districts. Schwaz is indicated by the red line and the remaining Austrian districts are represented by the solid grey lines. The dashed line represents the average 7-day incidence of all Austrian districts excluding Schwaz. Panel **b** shows the pre-trend for the municipalities along the border. Municipalities of Schwaz are in red, the ones of Kufstein are in orange and the ones of Innsbruck-Land are in blue. The observational (pre-treatment) period starts on the 1st of January 2021 and ends at the 12th of March 2021 (one day after the first dose of the vaccination campaign in Schwaz was administered).

design with bordering municipalities), as data on vaccination coverage for municipalities are also unavailable within the observational period.

Discussing the inclusion of vaccination rates in the SC framework, one should distinguish between two aspects. First, the choice of the control units which is determined by the weighting procedure, and, second, the effects of vaccination on infection rates after the treatment in both the treatment and the control group. For the former, it is crucial to mimic the pre-trend of the control group with respect to the dependent variable as close as possible to that of the treatment group. **Table S1** in the supplement shows relatively small differences between pre-treatment infection rates in the treatment and the control group. Therefore, we would expect only a small gain from including vaccination data and the bias from leaving out these data should be negligible. In fact, **Figure 1** of the manuscript shows that vaccination levels of Schwaz and all other Tyrolean districts have been very similar before the campaign, which was probably true also for all other Austrian districts due to the uniform execution of the national vaccination plan. Again, this let us expect that leaving out vaccination data in the weighting procedure does not matter too much. Similar holds true for the event study, in which we observe insignificant differences between infection rates of the treatment and the control group in the pre-treatment period.

We think that the second aspect, i.e., the omission of vaccination data on estimating the post-treatment effect, might be more important. We agree with your interpretation of the TWFE results, suggesting that the vaccination effect in Schwaz attenuates with the increasing vaccination coverage in the control units. From this, we would expect that the point estimates reported in the paper represent a lower bound of the vaccination effect. Unfortunately, we are not able to calculate the exact difference due to data availability. However, we took up your suggestion and spent a few more words on the potential role of increasing vaccine uptake in the control units in the Results section as well as in **Section 2** of the Supplement.

Comment R1.4: *Besides suggesting a recent comprehensive paper on synthetic controls methods by Abadie (2021, Journal of Economic Literature, <https://doi.org/10.1257/jel.20191450>), I believe the study would benefit in terms of transparency if the following details/robustness checks were reported in the appendix:*

In the new version of the paper, we now refer to Abadie (2021) in the Methods section.

a. Show weighting of other districts composing the SC

Out of all 92 remaining Austrian districts, the synthetic control group is composed by the districts of Hartberg-Fürstenfeld (24.1%), Hermagor (10.6%), Liezen (0.5%), Reutte (63.8%) and Steyr Stadt (1.1%). Notice that none of the chosen donors share a common border with Schwaz, making spillover effects due to the vaccination campaign unlikely. The composition of the control group is now mentioned in the Method section of the paper.

b. Test robustness of results to exclusion of some districts from the SC (see Abadie, 2021)

The requested robustness results are reported in **Figure R1-2** and are now included as **Figure S3** in the Supplement of the paper. As expected, we can see that leaving out one of the chosen donors in the baseline control group (blue dashed line) one at a time increases or lowers the number of daily infections in the synthetic control group, but only to a small

degree. Compared to the different 'leave-out' scenarios, Schwaz always had a considerably lower number of infections at the end of the study period. In the paper, we reported 2,460 cumulative infections for the control group (and 1,510 cases for Schwaz) at the end of the observational period (i.e., 19 weeks after the first dose), which decreases to 2,160 infections in the ('worst') scenario leaving out Reutte from the baseline control group. Thus, the difference between Schwaz and the control group is still substantial, letting us conclude that our main result about the effectiveness of the vaccination campaign seems robust.

Figure R.1-2: Treatment effect with one-leave-out

Notes: The figure depicts cumulative daily infections (per 100,000) for Schwaz (solid red line) and the synthetic control group (dashed blue line). The blue dashed line corresponds to the baseline control group reported in the paper. The grey lines represent the results of a synthetic control group where one of the chosen donors in the baseline are left out one at a time. The horizontal axis indicates the number of days relative to vaccination campaign (dose 1, indicated by “d1”). The pre-treatment period started 21 days (three weeks) before the first dose, the post-treatment period ended 133 days (19 weeks) after the first dose. The vertical dashed lines represent the first dose (d1) and the second dose (d2) administered in the vaccination campaign. Data: COVID-19 Open Data Portal (AGES/EMS).

c. Show (parallel?) pre-trends between treatment and control group for DID design (neighbouring districts)

See **Figure R.1-1 b** and the corresponding comments above. The figure shows parallel pre-trends between Schwaz and the neighbouring municipalities, which is an identifying assumption of the DID estimator. We also included this figure in the Supplement.

d. Show a full balance table of pre-treatment characteristics (controls vs. Schwaz) for both SC and DID (balance tables along the main set of district characteristics could go in the main

text)

Many thanks for this valuable comment. We address this comment separately for each methodological approach we used in our study. With regard to the SC-approach, **Table S1** in the Supplement presents a balancing table for the variables used in the weighting algorithm, which includes population, a district’s area, the number of municipalities within a district and, most importantly, the pre-treatment path of the outcome variable at different points in time equally spaced over the whole period before the vaccination campaign took place:¹ Day 2 (i.e., the first day after the beginning of the observational period), day 8, day 14, and 21 (i.e., the day before the vaccination campaign started on the 11th of March 2021). Further, we also report the root mean squared prediction error (RMSPE) informing how close the pre-trend of cumulative infections of the chosen donors is to the ones of Schwaz. Overall, we can see that the RMSPE is relatively small, letting us conclude that the SC-approach works well (which can be also inferred from the sheer graphical inspection of **Figure 2 a** in the manuscript).

With regard to the event study (ES/DID), our new **Table S2** in the Supplement provides a comparison of a set of characteristics for the municipalities in Schwaz and the ones along the border of the neighbouring districts of Kufstein and Innsbruck-Land. Notice that most of these variables are not available for all Austrian districts, which is the reason why we are not able to include them in the SC-approach.² The last column of **Table S2** shows the p-values of a (two-sided) t-test on the difference of these characteristics. It indicates that the treatment and control groups are comparable in many regards, suggesting a high validity of our ES approach.

1.2 Minor Comments

Comment R1.5: *Reporting the SC estimates in Table 1, alongside the two-period DID, would make the paper easier to read.*

Following your suggestion we included a new table to the manuscript, depicting the percent reduction estimated using the SC method alongside the estimated reduction based on the two-period DID (see **Table 2** of the manuscript). Comparing the numbers shown in the table suggests that the reductions in the medical outcomes are often of similar magnitude, regardless of the method we used. However, a few things need to be kept in mind when examining the numbers shown in the table. First, the numbers are based on different control regions, namely the border municipalities

¹Doing so, we follow recent research by Kaul et al. (2021) showing that using the full pre-treatment trajectory of the outcome variable as predictor would lead to potentially biased estimation results as compared to including only a limited number of outcome lags along with other covariates. In our application, it seemed reasonable to use not more than four outcome observations and to spread them evenly over the entire pre-treatment period. It should be noticed, however, that our estimation results did not change substantially when using five or six pre-treatment outcomes as predictors.

²In addition, we would not expect significantly different estimation results in the SC analysis, as the lags of the outcome variable generally pick up most of the variation in its pre-treatment path. As mentioned in footnote 1, including additional covariates (such as population, area and number of municipalities in our case) mainly has positive statistical properties and, therefore, avoids to obtain potentially biased estimation results.

(in case of the event-study) and the control districts (in case of the synthetic control group). Furthermore, please note that we do not have sequencing data on variants from districts/municipalities outside of Tyrol, which is the reason why we did not report them in this comparison table.

Comment R1.6: *Although the authors make the observational nature of the study clear, it would help to know something about how people were recruited for vaccination.*

Please see also our response to R1.2. Every citizen at the age of 16 or above who had his or her main legal residence in the district *prior* the announcement of the campaign (3rd of March 2021) was invited to get vaccinated at one of the district’s vaccination centres. The vaccination campaign was advertised in local media such as newspapers and radio stations. The reasonably high vaccination coverage of around 70% after the campaign suggests that a broad range of the general population participated in the campaign. We explain the recruitment of people in more detail now in **Section 2** of the Supplement.

Comment R1.7: *The authors may want to consider contrasting their estimates of the effect of a mass vaccination campaign with comparable studies on the effectiveness of other non-pharmaceutical public health interventions, for example, lockdowns.*

We think a very good comparison with another non-pharmaceutical intervention Austria used during the pandemic is the SARS-CoV-2 test requirement for individuals crossing the border of high-incidence regions. Five districts besides Schwaz had such a test requirement in place for some time. None of the five districts saw a drop in infections following the test requirement in any comparable way as we observed it in Schwaz after the vaccination campaign (see **Figure S7** in the Supplement as well as our explanations in the discussion section).

Regarding the effectiveness of lockdowns we do not know of any study quantifying its causal effect on hospitalisations or ICU cases for Austria. However, there exist studies which aimed to quantify this for other countries. For instance, Juranek and Zoutman (2021) estimated the effect of the lockdown in Denmark relative to the less strict non-pharmaceutical intervention Sweden pursued in March 2020 regarding the demand for health care. Specifically, they showed that cumulative hospitalisations were reduced to 3,878 patient-hospital-days, compared to a counterfactual scenario (with no strict lockdown) of 16,829. This means that the Danish lockdown led to 12,951 avoided patient-hospital-days, which is a reduction of $12,951/16,829 = 77\%$. Thus, the strict lockdown pursued by Denmark reduced hospitalizations by a larger percentage compared to the mass vaccination campaign (-36% and -43% depending on the method used, see **Table 2** of the manuscript).

Comment R1.8: *The discussion should highlight better the limited external validity of the results to other different settings.*

In the new version of the paper, we discuss limiting factors regarding the external validity of our study (see the Discussion section).

1.3 References

- Kaul, A., S. Klössner, G. Pfeifer and M. Schiefer, 2021. Never use all pre-intervention outcomes together with covariates. Working Paper, <https://mpra.ub.uni-muenchen.de/83790/>
- Juranek S. and F. Zoutman. 2021. The effect of non-pharmaceutical interventions on the demand for health care and on mortality: Evidence from COVID-19 in Scandinavia. *Journal of Population Economics* 34: 1299–1320.
- Shabir A. et al. 2021. Efficacy of the ChAdOx1 nCoV-19 Covid-19 Vaccine against the B.1.351 Variant. *New England Journal of Medicine* 384: 1885–1898.

2 Response to Referee (R2)

We are grateful for your excellent comments and suggestions. Below, we describe step-by-step how we have incorporated them.

2.1 Main Comments

Comment R2.1: *I would like to see a descriptive plot of the epidemic curve on the incidence in addition to the one on the cumulative scale. This could replace panel b of Figure 2 which shows information already present in panel a of the same figure. For Fig 4 it could be added to the supplement.*

Figure R.2-1 below shows the epidemic curve on the incidence in Schwaz relative to the synthetic control group. As requested, we used the 7-day incidence (per 100,000) of daily infections as the outcome variable. Two things are worth noting when looking at the figure. First, the treatment and the (synthetic) control group had very similar incidence rates in SARS-CoV-2 infections prior to the mass vaccination campaign. Second, although infections in Schwaz increased somewhat sharper than in the control group in the first days after the first dose, infection dynamics started to diverge around 2-3 weeks later. This is exactly the time period after which first effects of BNT162b2 materialized in the original phase 2/3 clinical trial (see Pollack et al. 2020: Figure 3). While there is also a small dip for the control group around that time (which was similar for the rest of Austria and related to limited testing due to the Easter Holidays), Schwaz followed its distinct and substantial decline in cases afterwards. In contrast, the control group witnessed a sharp increase with high incidence rates throughout April, which only began to reverse once a general trend of decreasing infection levels across Europe started in spring 2021. What we can also see from the figure is that the 7-day incidence rate is picking up more noise compared to the cumulative measure. In the manuscript, we included **Figure R.2-1** in the main text of the paper, see the new **Figure 2b**.

As suggested by the referee, **Figure S5** in the Supplement we now also provide plots of weekly incidence rates for hospitalizations and ICUs, corresponding to the cumulative measure shown in **Figure 3** (the previous Figure 4) of the manuscript. As can be seen from this figure, there is a break in the common trend between Schwaz and the synthetic control group after the vaccination campaign, although sometimes rather noisy. Especially for the weekly case incidence of ICU we are confronted with very low numbers. For instance, since the start of the vaccination campaign we only had a total number of 11 ICU admission in Schwaz and 24.9 ICU admissions in the synthetic control group (the corresponding numbers for general admissions are 39 and 65.7, respectively). This is why we think the cumulative measure is more informative here.

Comment R2.2: *I have only superficial knowledge on the synthetic control method, but to me it does not become clear how the control was actually constructed. I am sure there are numerous ways of doing this and the authors should clearly state what was done:*

Figure R.2-1: 7-day incidence in Schwaz and the synthetic control districts

Notes: The figure shows 7-day incidence of infections (per 100,000) in Schwaz (solid red line) and the synthetic control group (dashed blue line). The horizontal axis indicates the number of days relative to vaccination campaign (dose 1, indicated by “d1”). The pre-treatment period started 21 days (three weeks) before the first dose, the post-treatment period ended 133 days (19 weeks) after the first dose. Data: COVID-19 Open Data Portal (AGES/EMS).

- *Please specify which measure of similarity between Schwaz and the weighted control is being optimized. After a bit of research I concluded that it is likely a weighted sum of quadratic deviations across the covariates used for matching. How were these weights chosen? As the different covariates live on very different scales and orders of magnitude this seems like an important question.*

We select the weights to mimic the infection dynamics of Schwaz during the period before the intervention (i.e., the start of the vaccination campaign) as closely as possible. In our case, the pre-treatment period lasts 21 days. The number of donor districts is 92 (all Austrian districts excluding Schwaz and Rust, which includes about 2,000 inhabitants and is by far the smallest of all Austrian districts) and for which information on infections appear less reliable). Regarding the pre-treatment dynamics, we followed a recent paper by Kaul et al. (2021), who have shown that other potentially influential covariates become irrelevant if only the full pre-intervention entries of the outcome variable are used as predictors, which in turn might induce biased estimation SC results. For this reason, we divided the pre-treatment period into four periods with a length of one week and selected four time points accordingly: day 2 (one day after the begin of our observational period on the 18th of February 2021), day 8, day 14, and day 21 (one day before dose 1 of the vaccination campaign). In addition, we included

three covariates as optimizing variables: a district’s population, its area and its number of municipalities. In sum, we used 7 observations for Schwaz and each of the 92 districts to construct the synthetic control unit.¹ You are right to assume that the weighting procedure (described in footnote 1) basically minimizes a weighted sum of squared deviations. It is also true that the variables used in the weighting procedure might have different scales and magnitudes, but this should neither affect our chosen weights nor the subsequent estimation results. We checked this statement empirically by changing arbitrarily the scales of our three covariates (i.e., by multiplying them with a fixed scalar). It turns out that the chosen donors as well as their weights are the same as the ones reported in **Table S1**, and also the subsequent SC results of **Figure 2a** in the paper remained unchanged.

- *From the GitHub repository I can see that an implementation in Stata (apparently with default settings) was used to create the synthetic control. Please reference this in the text and provide a reference to the exact underlying method.*

In the Methods section of the paper, we now name the exact software (STATA MP version 16) and the package (Synth) used to execute the SC analysis. We also provide a reference to Abadie, Diamond and Hainmueller (2013), which described the exact method of the Synth package using R (the same authors also provided the corresponding Stata package).

Comment R2.3: *I do not understand regression equation (1) nor which data were included to fit the model:*

- *Why are both of the time indices t and w needed? I think only w should be used. The fact that t is also used makes me suspect that the authors fitted the model using all seven 7-day moving averages of each week. I don’t think this makes sense as these observations are not independent and essentially each independent data point will be used seven times. This will in turn lead to too small uncertainty intervals. Only one 7-day value per week should be included. Alternatively, seven one-day values could be included, but these are likely too noisy.*

In the previous version, we used daily infection data but used week fixed effects in the ES regressions. We agree that this might be somewhat confusing for the reader and, therefore, we followed your suggestion to switch from daily to the week level (i.e., one 7-day value per week calculated as the sum of infections over all weekdays). **Figure 4** in the manuscript shows the updated results. Furthermore, we also changed to this new weekly calculation of infection cases in **Table 1** (DID), so that the numbers shown in **Figure 4** and **Table 1** of the manuscript are comparable. To be consistent with **Figure 4b** we now also study the weekly number of confirmed VoCs pooled together (see column (2) of **Table 1**). This seems

¹Technically, let \mathbf{X}_0 denote a 7×1 vector of observations in Schwaz, \mathbf{X}_1 a 7×92 matrix with observations of the donor districts and \mathbf{w} a 7×1 vector of district weights, the synthetic control unit is defined by \mathbf{w}^* that optimizes the mean squared error $(\mathbf{X}_0 - \mathbf{X}_1 \mathbf{w})' \mathbf{V} (\mathbf{X}_0 - \mathbf{X}_1 \mathbf{w})$ s.t. $w_i \geq 0 \forall i = 1, \dots, 92$ and $\sum_i w_i = 1$. \mathbf{V} is a 7×7 symmetric and positive semi-definite matrix assigning different relevance to the characteristics in \mathbf{X}_0 and \mathbf{X}_1 . In our case, we used a Stata-package to implement the weighting procedure (see next comment) which chooses the diagonal matrix \mathbf{V} such that the root mean squared prediction error (RMSPE) of the outcome variable is minimized in the pre-treatment period (see also Abadie, Diamond and Hainmueller 2010: 496).

reasonable given the fact that the delta variant entered Austria only at the end of our study period, confronting us with very low or zero number of cases in some municipalities. This is also why we decided to interpret our results a bit more cautiously regarding variant cases.

- *How can the term $D_{it,w}$ enter 22 times as a covariate? I assume that something went wrong in the notation as this cannot be what was actually done. As is, equation (1) could be re-written as*

$$y_{it,w} = \delta_i + \delta_w + \left(\sum_{k=-6}^{-1} \beta_k + \sum_{k=1}^{16} \beta_k \right) \times D_{it,w},$$

which is not meaningful. My understanding from looking at the code (I am not a Stata user) and the verbal description is that the authors did something like

$$y_{i,w} = \delta_i + \delta_w + \sum_{k=-6}^{-1} \beta_k D_{i,w-k} + \sum_{k=1}^{16} \beta_k D_{i,w-k},$$

where $D_{i,w-k} = 1$ if and only if (i is a municipality which received the treatment and $w - k = 0$), i.e. the observation comes from Schwaz and is from k weeks after the start of the intervention (which is at week 0).

Many thanks for this comment. In the equation, we intended to show that the observational time unit is a day which is nested in a week and captured by the week fixed effects. As mentioned in the previous comment, we agree that this is somewhat misleading. Hence, we now changed to the week level, making it easier to write down our estimation equation. In fact, it is now more or less equivalent to your second suggestion, and reads as

$$y_{i,w} = \delta_i + \delta_w + \sum_{k=-6}^{-1} \beta_k D_{i,w+k} + \sum_{k=1}^{19} \beta_k D_{i,w+k} + \varepsilon_{i,w}.$$

The difference to your suggestion is the subscript of D in the summation operator, where we use $w + k$ instead of $w - k$. Accordingly, the first expression including $D_{i,w+k}$ relates to the pre-treatment period, and the second one to the post-treatment period. Further, $D = 1$ if a municipality received at treatment (i.e., lies within the district of Schwaz) and $w \geq 6$. We changed the equation in the manuscript accordingly.

Comment R2.4: *A general question which concerns both equation (1) and the difference in difference approach: I wonder whether this "linear" approach is the most meaningful given the exponential nature of an epidemic. As an example: Consider two otherwise comparable regions A and B which at time 1 have seven-day incidences of 50 and 75, respectively. After an intervention in region B, at time 2 they have 100 and 130, respectively. In your DID setting, the effect of the intervention would be quantified as $(130 - 75) - (100 - 50) = 5$, i.e. a slight worsening of the situation. I would, however, argue that the intervention had a desirable impact given that cases only rose by 73% rather than 100% as in region A. I suggest to re-run the regression model and difference in difference computation using log-transformed incidence values of $z_{iw} = \log(y_{i,w} + 1)$, so that the*

exp-transformed parameters β_k have a multiplicative interpretation. I assume results to be robust to this, but I think it would make sense to report both versions (one could be moved to the supplement).

We fully agree with your concern and are, at the same time, sorry that we did not better describe the way in which we estimated our percentage impact in the DID approach. In fact, we ran a regression using the log-transformed outcome variable. The only difference to your suggestion is that we did not use $z_{iw} = \log(y_{iw} + 1)$ but rather the hyperbolic sine transformation of the form $z_{iw} = \ln \left[y_{iw} + \sqrt{(y_{iw}^2 + 1)} \right]$, which has been shown a more recent but also widely accepted approach of dealing with the $\ln(0)$ -problem in a regression framework (see, e.g., MacKinnon and Magee 1990). In any case, running a regression with a log-transformed dependent variable and an independent variable in levels (as in our case) delivers a so-called semi-elasticity, which can be interpreted as percentage impact after transforming the regression coefficients according to $(e^\beta - 1) \times 100$, where e is Euler's number and β is the parameter estimate of the DID interaction term. In the new **Table 2** of the paper, we now report percentage values that are exactly calculated in this way.

Comment R2.5: *In Figure 3 I find it difficult to compare the reductions in the different age groups as I am unaware of the general levels of incidence in the different age groups. Maybe the differences look bigger for some age groups because the numbers are generally larger in that age group. I suggest to split the figure into two panels, one showing the current figure and one showing the relative difference in cumulative incidence since the intervention (i.e. ignoring everything that happened prior to the intervention).*

Many thanks for this comment. Please also consider our response to your comment R2.7 below, which is related to this concern as it also belongs to the interpretation of our SC results. We followed your recommendation and plotted a figure with two panels (see **Figure R2-2** below): Panel **a** shows our original figure and panel **b** shows relative differences in cumulative incidences since the intervention. Two conclusions can be drawn from the figure: First, the differences between the age groups are no longer as strong as in the figure with absolute differences. Therefore, we decided to discuss the age-specific profiles in the paper more carefully. Second, the relative differences in cumulative infections are somewhat noisy in the first days after the first dose. In particular, the interpretation of the relative effects also depends on the starting point at which we calculated the overall effect of the vaccination campaign. For instance, starting 14 days after the roll-out of the first dose (as in the DID estimation) provides different results to the ones of choosing dose 1 as the starting point (compare **Figure R2-2 c** and **Figure R2-2 b**). However, although **Figure R2-2** provides an unclear picture of the relative vaccination effects, it is obvious that the age specific differences become much smaller when interpreting relative rather than absolute changes. Therefore, we decided to interpret the age effects of the vaccination campaign more cautiously and propose to shift our original graph to **Figure S4** of the Supplement. There, we also plotted the cumulative infections of Schwaz and the synthetic control group for each age group separately, which we think is the most transparent way to show the differing levels of the vaccination effects. We hope that you are satisfied with our preferred solution. However, if you prefer one of the panels in **Figure R2-2**, it is no problem to replace **Figure S4** by one of the versions shown in **Figure**

R2-2.

Figure R.2-2: Age-specific differences in cumulative daily infections between Schwaz and the synthetic control group

Notes: The figure plots the difference in cumulative daily infections (per 100,000) between Schwaz and the synthetic control group for each age cohort in the sample. **a** relates to the absolute, and **b** to the relative (percentage) difference within each age group starting with the roll-out of the first dose. **c** and **d** refer to different starting points (i.e., 14 days after the first dose and the day where the roll-out of the second dose was started). A positive difference indicates higher infection rates for the control group than for Schwaz. The horizontal axis indicates the number of days relative to vaccination campaign (dose 1). The pre-treatment period started 21 days (three weeks) before the first dose, the post-treatment period ended 133 days (19 weeks) after the first dose. The vertical dashed lines represent the first dose (d1) and the second dose (d2) administered in the vaccination campaign.

Comment R2.6: *Please provide a reference for the "standard permutation test" mentioned in several instances.*

We applied a permutation test as proposed by Abadie, Diamond and Hainmueller (2010), which basically uses an exact Fisher test and is recently described by Abadie (2021: 403). The new references are included in the manuscript and in the Supplement.

Comment R2.7: *I am puzzled by the percentage numbers reported for the reduction of cases, hospitalizations etc on page 5:*

- *Line 195: 900 avoided infections compared to 2400 total infections is a 37% reduction. 900 avoided infections compared to roughly 1200 total infections since the start of the treatment is a 75% reduction. Where do the 53.6% come from? What am I missing?*
- *Line 225: 55.7 avoided hospitalizations for 126.8 in the untreated group is a reduction of 43%. 55.7 is 78% of 71.0 (i.e. the number in the treated group). So it would be correct to say that in the unvaccinated group hospitalization was 78% higher, but the current statement is incorrect.*
- *The same problem is present in line 232 (5.2 is 31% of 16.6, not 21.8).*
- *Please carefully check all reported percentage values and be sure that they really correspond to the verbal interpretation provided!*

We are grateful for this comment. Obviously, we provided a somewhat confusing interpretation of our results. With respect to your example given in the first bullet point, we would prefer the first interpretation, which is that the vaccination campaign resulted in a 37% decrease of infections in Schwaz. In the manuscript, we now checked all numbers again and corrected the corresponding percentage values throughout.

Comment R2.8: *I have some questions with respect to confounding:*

- *The setting in Schwaz is probably as close to an experimental design as one can get in an epidemiological study of mass vaccination. However, as the authors correctly point out, this remains an observational study with a risk of confounding. A possible difficulty I see is the following: By announcing the mass vaccination programme, a strong signal was sent to the population that the situation was drastic and possibly out of hand. Could that have led to adaptations in behaviour which decreased transmission?*

You are right that the announcement of the campaign may have triggered behavioural changes that influenced virus transmission. In this respect, it is important to note that the mass vaccination campaign was announced on the 3rd of March 2021, which is probably the time when the strongest signal about an extraordinary situation was sent to the population. In our analysis we observed case numbers to drop significantly only after the second dose (8th to 11th April), which was many weeks after the announcement of the campaign. In addition,

it is important to mention that the reason of the vaccination campaign was not so much extreme levels of infection spread in Schwaz (see **Figure S1a** in the Supplement) but rather the occurrence of the B.1.351 variant. In February 2021, a local outbreak of B.1.351 in the district of Schwaz happened, and the concern of the government and its expert panel was that the spike protein mutations of this variant may make it less susceptible to the immune response induced by vaccines. This is why authorities decided for the local mass vaccination campaign in order to create a "vaccination ring" around this outbreak of B.1.351. Thus, the main reason was not so much extreme levels of infections with a high risk for the local population to get infected (which may have triggered strong behavioural changes) but rather the occurrence of this specific variant. Finally, we analysed Google mobility data and found no reduction in mobility in Schwaz relative to the synthetic control group, which would be a sign of changes in people's behaviour (see Supplementary **Figure S6**). While we cannot rule out behavioural changes we think the overall pattern of our findings with a significant drop in cases exactly after the second dose of the campaign suggest that this reduction was driven by the mass vaccination campaign. Nevertheless, in the Discussion section of our paper we now discuss changes in people's behaviour after the announcement as a potential confounder.

- *What made me wonder about this is the fact in the left panel of Fig 5 a rather pronounced effect (though not statistically significant) is visible already one week after the intervention. According to e.g. the Australian Department of Health, "partial protection against COVID-19 may be as soon as 12 days after the first dose, this protection is likely to be short lived" (<https://www.health.gov.au/initiatives-and-programs/Covid-19-vaccines/is-it-true/is-it-true/-how-long-does-it-take-to-have-immunity-after-vaccination>). I am not an immunologist, but would not expect to see an effect one week after the first vaccination. Of course one should not overinterpret estimates with wide error bars, but I would still like to ask the authors for a brief comment on this aspect.*

Unfortunately, the estimates based on the border design carry some noise due to the fact that they are based on a limited sample of bordering (often small) municipalities just outside of the district of Schwaz. This is probably why we observe some fluctuation in the weekly coefficients. For instance after the described drop in week 1 after the first dose, we observe a small rise in cases again in week 2 (both not statistically significant). However, a much more sustained and also significant drop materializes later, right after the second dose. This is also around the time when the vaccine should take its full effect. This is confirmed by our estimates using the SC method, which is based on a larger population (namely districts) and therefore less noisy.

- *That being said, I do not think that changes in population behaviour are relevant for the observed changes in the medium term, which are the most relevant for the overall results and interpretation.*

We agree with the referee that taking together all presented evidence, our findings suggest that the mass vaccination campaign was indeed the main driver of the large reduction of

infections in Schwaz relative to the control regions.

- *Another difference between Schwaz and the synthetic control (or any point of comparison in the study) is that Schwaz had "one of the largest outbreaks of B.1.351 and B.1.1.7/E484K in Europe". If at all, this should bias results towards a lower effect of mass vaccination, so this is not problematic. But I would still be interested in a brief comment on whether this outbreak must be assumed to have a relevant impact on the subsequent course of the epidemic in Schwaz.*

The outbreak of variant cases in Schwaz was not so exceptional regarding incidence rates. For instance, other districts in Austria had higher rates of infections over the course of the pandemic, for instance during winter 2021 (see here also our explanations and figures provided in response to the referees' comment R1.2). What the government and its expert panel was worried about was the cluster of variant cases. As documented by our study, the variants were successfully contained by the mass vaccination campaign (see **Figure 4b** of the paper). However, similar to almost all regions in Europe, the delta variant is now the main SARS-CoV-2 strain present in Schwaz.

Comment R2.9: *Supplementary Figure 1 does not seem to be referenced in the main text. I think this is an interesting figure as it illustrates that the synthetic control approach works "well enough" and the observed effect is unlikely to be due to chance alone.*

In the new version of the paper we mention and reference the previous **Figure 1** of the Supplement, which is now **Figure S2** in the Supplement. Further, we included a new **Figure S3** in the Supplement, showing an additional robustness as requested by Referee 1. In particular, we left out one of the chosen donors one at a time and applied our SC procedure on the remaining donors to see whether our results are influenced by one particular donor. **Figure S3** shows that this is certainly not the case, giving a further hint that the SC approach works well. Both figures are now mentioned in lines 104 and 325/327 of the manuscript.

Comment R2.10: *The GitHub repository referenced by the authors contains analysis codes, but no data to replicate the presented analyses. I strongly encourage making all materials available in a self-contained and stable repository so the results can be replicated.*

Unfortunately, we are not able to share this data freely. The data we use are collected by the Austrian National Public Health Institute (Gesundheit Österreich GmbH, GÖG), and is provided to us through a restricted-access agreement. Access to this dataset can be given to other researchers through direct application for data access to the GÖG. The sequencing and vaccination data is also non-public, but can be obtained by sending an email inquiry to the "Amt der Tiroler Landesregierung" (lwz@tirol.gv.at). We are very sorry that we are unable to post this data on GitHub, but we do not want to risk a breach of the agreements we signed.

2.2 Minor Comments

Comment R2.11: Please add the word "cumulative" to the axis labels in Fig 4.

This is changed now (**Figure 4** in the manuscript).

Comment R2.12: I suggest to use different colours for curves referring to the whole of Tyrol and to the synthetic control (these are currently both shown in blue)

Throughout the paper, we now use a solid red line to indicate Schwaz and a dashed blue line to indicate the respective control group (particularly in the SC analysis). As suggested, the whole of Tyrol we now indicated in orange (see **Figure 1** in the manuscript and **Figure S1b** in the Supplement).

2.3 References

- Abadie A., A. Diamond and J. Hainmueller, 2010. Synthetic control methods for comparative case studies: Estimating the effect of California's tobacco control program. *Journal of the American Statistical Association* 105, 493–505
- Abadie A., A. Diamond and J. Hainmueller, 2013. Synth: An R Package for Synthetic Control Methods in Comparative Case Studies. *Journal of Statistical Software* 42 (open access)
- Abadie A., 2021. Using synthetic controls: Feasibility, data requirements, and methodological aspects. *Journal of Economic Literature* 59, 391–425
- Kaul, A., S. Klössner, G. Pfeifer and M. Schiefer, 2021. Never use all pre-intervention outcomes together with covariates. Working Paper, <https://mpra.ub.uni-muenchen.de/83790/>
- MacKinnon, J.A. and L. Magee, 1990. Transforming the dependent variable in regression models. *International Economic Review* 31, 315–339.
- Polack F.P., S.J., Thomas, B. Kitchin et al. 2020. Safety and Efficacy of the BNT162b2 mRNA Covid-19 Vaccine. *New England Journal of Medicine* 383, 2603–2615.

REVIEWERS' COMMENTS

Reviewer #1 (Remarks to the Author):

Review report on "The effects of rapid mass vaccination against SARS-CoV-2: Quasiexperimental evidence from an early VoC hotspot"

I am thankful to the authors for the detailed rebuttal letter and for the revised version of the manuscript. The results are now much more convincing, better supported by an extensive battery of

robustness checks, and transparently reported. With all due precautions related to the use of observational studies to draw causal interpretation, this represents an important piece of evidence that

makes the most out of a quasi-experiment around vaccination, given the data at hand.

I only have a few minor comments, mostly suggestions to improve clarity and readability. I appreciate

that the authors may have a different (and possibly better) taste in addressing these points, of course.

1. Page 3 (lines 106-108): The following sentence is unclear: "Relating the observed difference of 959 (avoided) infections to the number of infections in the synthetic control group (2,460) gives a reduction of 38.8%". I suggest rephrasing to something like "The difference in infections between the treatment municipalities and the synthetic control can be interpreted as estimate of avoided infections. Our results suggest that the mass vaccination campaign resulted in 959 avoided infections in the four months after the first dose of vaccine. This is equivalent to a reduction of 38.8% compared to the synthetic control."

2. On the same point above, why didn't you compute 95% confidence intervals for the number of avoided infections (and perc. reduction)? I would assume that a bootstrapping procedure would allow you to obtain them.

3. Page 4 (lines 173-175): "... we used a log-transformed regression and received a semielasticity (see also the Methods section)." I suggest changing to "... we used a logtransformed regression and obtained semi-elasticities (see also the Methods section)."

4. Related to points 2 and 3 above, I suggest reporting all results in Table 2 with the corresponding confidence intervals (which you have for the 2-period DID, and can compute for the difference in cumulative infections between treatment and synthetic control). This would make clearer that the numbers result directly from your empirical strategy and are not just back of the envelope calculations. Personally, I'd suggest also moving the current Table 1 to supplementary materials, reporting the VoC result in the same table (currently Table 2) although only for the ES for this outcome.

5. Page 6 (line 238): Perhaps you want to consider changing to something like "It is important to stress that our estimates ...".

6. Page 8 (lines 352-356): I see what you mean, but the term "leads" may read a bit counterintuitive when considering sums of $k = -6$ to -1 in your specification (1). Perhaps you could rephrase to something like "anticipated" and "lagged" treatment effects? This is a very minor point.

7. Page 9 (lines 381-383): This sentence needs revision: "On the other hand, the SC method using only districts which are much further apart from Schwarz rules out such cross-protection effects, but does not share some local geographical characteristics which may affect infection spread."

8. Figure 4: Add a heading or some other way (e.g. adding different titles to the y-axis) to differentiate content of plots a (all infections) and plot b (only VoC). I appreciate that this is all written in the manuscript, but the reader should now what's plotted looking at the figure alone.

9. Abstract: at the risk of sounding pedantic, I invite you to consider toning down the last sentence "Our results demonstrate that rapid mass vaccination is an effective tool to curb the spread of SARS-CoV-2.". Further in the paper you use "suggest", which I believe is more appropriate. An alternative would be something like "Our results support the idea that rapid mass vaccination campaigns are an effective tool to curb the spread of SARS-CoV-2."

Reviewer #2 (Remarks to the Author):

The revised version of this very interesting paper has been prepared carefully and I am happy to recommend it for publication. I have a few additional remarks, but these are only intended to help make the manuscript a bit easier to read for an epidemiology/biostatistics audience.

1. Following up on my previous comment on the SC method: This part is now better documented and contains relevant references. However, I think two more sentences could be spent on the following to make clear what is happening and avoid scepticism from researchers unfamiliar with this method:

1.1 Please explicitly state that the synthetic control group is constructed as a convex combination of donors by minimizing a weighted sum of squared deviations for the matching variables. Weights are chosen in a data-driven way (I think it is a sort of cross validation - if it is this may be the most straightforward way to get the idea across).

1.2 Make a brief statement on identifiability as to most statisticians it will come unexpected that weights for 92 donors are estimated based on matching just seven data points. In my understanding identifiability is ensured by the convexity constraint (i.e. the constraint of non-negative weights summing to one). It may be helpful to refer to footnotes 18 and 19 from <https://inferenceproject.yale.edu/sites/default/files/jel.20191450.pdf>

2. The paper uses some economics jargon which I think may make it a bit more difficult to read for an epidemiology audience:

2.1 I am sorry about the confusion concerning the log transformation in my previous review. The term semi-elasticity is probably obvious to an economics audience, but to my knowledge unusual in epidemiology (I confirmed this with several colleagues). I suggest to add the regression equation with log transformation used here, which will make this point straightforward to understand for everybody. In this context, a one-sentence description of "semi-elasticity" could be given (the concept being simple, but the term possibly unfamiliar to many readers).

2.2 The same holds for the term "event study". From the context it is kind of clear what this means, but few epidemiologists will know that term. Adding a brief verbal description and possibly a standard reference could help here (along the lines of: a method commonly used in econometrics to assess the effect of an [external] event on a quantity of interest, e.g. a stock prize. This at least was my understanding).

3. Figure S5 is not referenced in the text.

4. I suggest to use Greek letter names ("alpha", "beta", "delta") instead of or in addition to the B.<...> names of variants where they exist. E.g. at the bottom of page 5 both nomenclatures are mixed and personally I find the Greek letters easier to read and remember (keep in mind that this text should also be readable many years from now).

5. Could the authors perform the same permutation test as for cases also for hospitalizations and ICU need?

6. What does the row "observations" in Table 1 stand for?

7. Add confidence intervals to Table 2 where available (similar to Tab 1)

8. Around line 112 you could add that the population of Schwaz only had partial protection over an important part of the "study period" (or even none, as the cumulative numbers are counted from the beginning of January).

9. My understanding is that the same synthetic control group was used for cases, hospitalizations and ICUs. However, matching was based only on case numbers. Is that correct? If so I think this would be worth mentioning.

Minor / language:

- add "statistically" to the phrase "We found a significant reduction ... " in line 189. Same for line 185.

- "residing" in line 211 sounds strange to me. Replace by "located"?

- line 231 let -> led

- line 330 struy -> study?

- line 175: here the point estimate is reported as a positive number and the CI as two negative numbers, could be formulated more coherently

Comments on the final Revision of Manuscript NCOMMS-21-33616

The effects of rapid mass vaccination against SARS-CoV-2 and its Variants-of-Concern: Evidence from an early VoC hotspot

We thank the Editor for inviting us to do a final round of revisions to our manuscript. Again, we have received very helpful remarks from the two referees (henceforth R1 & R2). Again, we have done our best to address all suggestions and remarks, and we hope that our responses are sufficiently clear and comprehensible. Below, we address all the recommendations of the referees as they appear in the reports. All page numbers in our response letter refer to the revised version of the paper.

1 Response to Referee (R1)

We are grateful for your excellent comments and suggestions. Below, we describe step-by-step on how we have treated them in the manuscript.

1.1 Comments

Comment R1.1: *Page 3 (lines 106-108): The following sentence is unclear: "Relating the observed difference of 959 (avoided) infections to the number of infections in the synthetic control group (2,460) gives a reduction of 38.8%". I suggest rephrasing to something like "The difference in infections between the treatment municipalities and the synthetic control can be interpreted as estimate of avoided infections. Our results suggest that the mass vaccination campaign resulted in 959 avoided infections in the four months after the first dose of vaccine. This is equivalent to a reduction of 38.8% compared to the synthetic control."*

Many thanks for this recommendation, which in fact improves the interpretability of the numbers reported in the paper. We changed the respective sentence in the manuscript.

Comment R1.2: *On the same point above, why didn't you compute 95% confidence intervals for the number of avoided infections (and perc. reduction)? I would assume that a bootstrapping procedure would allow you to obtain them.*

Confidence intervals for SC estimates have only recently been discussed in the SC literature. In applied work, they are hardly reported yet. The reason is that they are based on standard errors drawn from a permutation test as we did in the previous version's Figure S2 of the Supplement (notice that bootstrapping is not proposed in this literature). Firpo and Possebom (2018) provide a step-by-step description on how to calculate confidence sets based on such a permutation test. We followed their guidance and calculated a confidence interval of $[-72.7; -1555.9]$ for our baseline SC estimate in Figure 2a of the manuscript (i.e., the 959 difference in avoided infections mentioned in your comment 1). Both confidence bounds are negative, indicating that the vaccination campaign contributed significantly to the observed reduction of infections in Schwaz. Notice, however, that

these confidence intervals have a very specific interpretation, i.e., one that relies on the relative position of the treated unit within the distribution of SC estimates in the whole group of placebo units. Further, the CIs we obtained from the Firpo-Possebom-procedure cannot directly be compared to the traditional 95%-CIs (although ours come close to such levels). Therefore, to avoid complicated explanations in the paper and to preclude any misinterpretations compared to the 95%-CI of our DID estimates, we decided to follow previous applied work and report only the point estimates of the SC method.

Comment R1.3: *Page 4 (lines 173-175): "... we used a log-transformed regression and received a semielasticity (see also the Methods section)." I suggest changing to "... we used a logtransformed regression and obtained semi-elasticities (see also the Methods section)."*

We changed the respective sentence in the manuscript as suggested by the referee.

Comment R1.4: *Related to points 2 and 3 above, I suggest reporting all results in Table 2 with the corresponding confidence intervals (which you have for the 2-period DID, and can compute for the difference in cumulative infections between treatment and synthetic control). This would make clearer that the numbers result directly from your empirical strategy and are not just back of the envelope calculations. Personally, I'd suggest also moving the current Table 1 to supplementary materials, reporting the VoC result in the same table (currently Table 2) although only for the ES for this outcome.*

See also our comment on your comment 2 from above. Further, we followed your recommendation and moved former Table 1 to the Supplement. There, it is Table 3 now.

Comment R1.5: *Page 6 (line 238): Perhaps you want to consider changing to something like "It is important to stress that our estimates .."*

We changed the respective sentence in the manuscript as suggested by the referee.

Comment R1.6: *Page 8 (lines 352-356): I see what you mean, but the term "leads" may read a bit counterintuitive when considering sums of k -6 to -1 in your specification (1). Perhaps you could rephrase to something like "anticipated" and "lagged" treatment effects? This is a very minor point.*

Although it is common in the ES-literature to talk about "leads" and "lags", we see your point and now write "... k in the sum operators indicate pre- (first sum) and post- (second sum) treatment effects."

Comment R1.6: *Page 9 (lines 381-383): This sentence needs revision: "On the other hand, the SC method using only districts which are much further apart from Schwaz rules out such cross-protection effects, but does not share some local geographical characteristics which may affect infection spread."*

We changed this sentence into two sentences: *"In contrast, the SC method relies only on districts much further apart from Schwaz, which rules out such cross-protection effects. However, these districts do not share some of the local geographical characteristics that potentially influence infection*

spread.”

Comment R1.7: *Figure 4: Add a heading or some other way (e.g. adding different titles to the y-axis) to differentiate content of plots a (all infections) and plot b (only VoC). I appreciate that this is all written in the manuscript, but the reader should now what’s plotted looking at the figure alone.*

Thank you very much for this comment. We changed the titles on the y-axis, see Figure 4 of the manuscript.

Comment R1.8: *Abstract: at the risk of sounding pedantic, I invite you to consider toning down the last sentence “Our results demonstrate that rapid mass vaccination is an effective tool to curb the spread of SARS-CoV-2.”. Further in the paper you use “suggest”, which I believe is more appropriate. An alternative would be something like “Our results support the idea that rapid mass vaccination campaigns are an effective tool to curb the spread of SARS-CoV-2.”*

We changed the abstract accordingly.

1.2 References

Firpo, S. and V. Possebom, 2018. Synthetic control method: Inference, sensitivity analysis and confidence sets. *Journal of Causal Inference* 6: 20160026.

2 Response to Referee (R2)

We are grateful for your excellent comments and suggestions. Below, we describe step-by-step how we have incorporated them.

2.1 Main Comments

Comment R2.1: *Following up on my previous comment on the SC method: This part is now better documented and contains relevant references. However, I think two more sentences could be spent on the following to make clear what is happening and avoid scepticism from researchers unfamiliar with this method:*

- *Please explicitly state that the synthetic control group is constructed as a convex combination of donors by minimizing a weighted sum of squared deviations for the matching variables. Weights are chosen in a data-driven way (I think it is a sort of cross validation - if it is this may be the most straightforward way to get the idea across).*

Thank you very much for this, we included a statement in the paper as suggested.

- *Make a brief statement on identifiability as to most statisticians it will come unexpected that weights for 92 donors are estimated based on matching just seven data points. In my understanding identifiability is ensured by the convexity constraint (i.e. the constraint of non-negative weights summing to one). It may be helpful to refer to footnotes 18 and 19 from <https://inferenceproject.yale.edu/sites/default/files/jel.20191450.pdf>*

Thanks again for this reference, which is now mentioned in the manuscript.

Comment R2.2: *The paper uses some economics jargon which I think may make it a bit more difficult to read for an epidemiology audience:*

- *I am sorry about the confusion concerning the log transformation in my previous review. The term semi-elasticity is probably obvious to an economics audience, but to my knowledge unusual in epidemiology (I confirmed this with several colleagues). I suggest to add the regression equation with log transformation used here, which will make this point straightforward to understand for everybody. In this context, a one-sentence description of "semi-elasticity" could be given (the concept being simple, but the term possibly unfamiliar to many readers).*

We followed your suggestion and added the regression equation we estimated to obtain the DID effect to the Methods section of the paper (see equation (2)). In addition, we now describe what the semi-elasticity measures: It measures the percentage change in the respective outcome (e.g., number of infections) in response to a change in another variable, in our case the binary interaction term of equation (2).

- *The same holds for the term "event study". From the context it is kind of clear what this means, but few epidemiologists will know that term. Adding a brief verbal description and*

possibly a standard reference could help here (along the lines of: a method commonly used in econometrics to assess the effect of an [external] event on a quantity of interest, e.g. a stock prize. This at least was my understanding).

In the new version of the paper we included a sentence with a verbal description along the lines you suggested. Doing so, please notice that the recent (micro-)econometric literature now has a different usage of the term "event study". In the past, an "event study" primarily meant the study of extraordinary stock returns due to an event (e.g., a monetary shock). Today, this term is simply used in the context of a two-way fixed effects model, which allows to figure out the dynamic causal effects of an intervention (in other words, it informs about the time pattern of the static two-period difference-in-difference estimate). The references we would like to mention in this context are Cunningham (2021) and, for a widely cited application in health economics, Dobkin et al. (2108). Both references are included in the manuscript.

Comment R2.3: *Figure S5 is not referenced in the text.*

In the new version of the paper we reference Figure S5.

Comment R2.4: *I suggest to use Greek letter names ("alpha", "beta", "delta") instead of or in addition to the $B < \dots >$ names of variants where they exist. E.g. at the bottom of page 5 both nomenclatures are mixed and personally I find the Greek letters easier to read and remember (keep in mind that this text should also be readable many years from now).*

Throughout the paper and supplement we changed to the greek letters.

Comment R2.5: *Could the authors perform the same permutation test as for cases also for hospitalizations and ICU need?*

See the new Figure 3 in the Supplement, which now provides the requested permutation tests for hospitalizations (panel a) and ICU admissions (panel b). Again, the permutation test seems to confirm that ICU number should be interpreted cautiously given the relatively low numbers of ICU admissions at the district level within our observational period.

Comment R2.6: *What does the row "observations" in Table 1 stand for?*

It stands for the number of observations in the regression analysis. For instance, in column (1) 1,323 belongs to the number of municipalities (=49) times the number of weeks (=27) in the study. In the new version of the paper (Table 3 in the Supplement, see next comment), we now write "*No. of observations*".

Comment R2.7: *Add confidence intervals to Table 2 where available (similar to Tab 1).*

First of all, notice that R1 wanted to move Table 1 of the manuscript to the appendix. We followed this suggestion, so that former Table 2 is now Table 1 in the manuscript. In this table, we now added the 95%-CI for the percentage changes of the DID estimates (reported in Table 3 of the Supplement).

Comment R2.8: *Around line 112 you could add that the population of Schwaz only had partial protection over an important part of the "study period" (or even none, as the cumulative numbers are counted from the beginning of January).*

We included the suggested sentence in the manuscript.

Comment R2.9: *My understanding is that the same synthetic control group was used for cases, hospitalizations and ICUs. However, matching was based only on case numbers. Is that correct? If so I think this would be worth mentioning.*

No, we applied our matching algorithm for each of the outcome variables separately, i.e., they are not only based on case numbers but different for case numbers, hospitalizations and ICUs. We decided to use separate donors for each outcome variable to ensure comparable pre-trends in each case. This is clarified now in the paper, i.e. each list of donors is mentioned in the notes of Figures 2 (cases) and 3 (hospitalizations).

2.2 Minor comments

- *add "statistically" to the phrase "We found a significant reduction ... " in line 189. Same for line 185. Done.*
- *residing" in line 211 sounds strange to me. Replace by "located"? We replaced it by "living".*
- *line 231 let → led. Done.*
- *line 330 struy → study? Done.*
- *line 175: here the point estimate is reported as a positive number and the CI as two negative numbers, could be formulated more coherently. Done.*